# Herbaceous perennial plants with short generation time have stronger responses to climate anomalies than those with longer generation time

Aldo Compagnoni [1,2✉], Sam Levin[1,2], Dylan Z. Childs [3], Stan Harpole [1,2,4], Maria Paniw[5], Gesa Römer[6,7], Jean H. Burns [8], Judy Che-Castaldo[9], Nadja Rüger[2,10,11], Georges Kunstler [12], Joanne M. Bennett [1,2,13], C. Ruth Archer[14,15], Owen R. Jones [6,7], Roberto Salguero-Gómez[16,18] & Tiffany M. Knight [1,2,17,18]

There is an urgent need to synthesize the state of our knowledge on plant responses to climate. The availability of open-access data provide opportunities to examine quantitative generalizations regarding which biomes and species are most responsive to climate drivers. Here, we synthesize time series of structured population models from 162 populations of 62 plants, mostly herbaceous species from temperate biomes, to link plant population growth rates ($\lambda$) to precipitation and temperature drivers. We expect: (1) more pronounced demographic responses to precipitation than temperature, especially in arid biomes; and (2) a higher climate sensitivity in short-lived rather than long-lived species. We find that precipitation anomalies have a nearly three-fold larger effect on $\lambda$ than temperature. Species with shorter generation time have much stronger absolute responses to climate anomalies. We conclude that key species-level traits can predict plant population responses to climate, and discuss the relevance of this generalization for conservation planning.

[1] Martin Luther University Halle-Wittenberg, Halle (Saale), Germany. [2] German Centre for Integrative Biodiversity Research (iDiv) Halle-Jena-Leipzig, Leipzig, Germany. [3] Department of Animal and Plant Sciences, University of Sheffield, Sheffield, UK. [4] Department of Physiological Diversity, Helmholtz-Centre for Environmental Research—UFZ, Leipzig, Germany. [5] Department of Evolutionary Biology and Environmental Studies, University of Zurich, Winterthurerstrasse 190, Zurich CH-8057, Switzerland. [6] Interdisciplinary Center on Population Dynamics, University of Southern Denmark, Odense M, Denmark. [7] Department of Biology, University of Southern Denmark, Odense M, Denmark. [8] Department of Biology, Case Western Reserve University, Cleveland, OH, USA. [9] Alexander Center for Applied Population Biology, Conservation & Science Department, Lincoln Park Zoo, Chicago, IL, USA. [10] Smithsonian Tropical Research Institute, Apartado, Balboa, Ancón, Panama. [11] Department of Economics, University of Leipzig, Leipzig, Germany. [12] Univ. Grenoble Alpes, INRAE, UR LESSEM, Grenoble, France. [13] Centre for Applied Water Science, Institute for Applied Ecology, The University of Canberra, Canberra, Australian Capital Territory, Canberra, Australia. [14] Centre for Ecology and Conservation, College of Life and Environmental Sciences, University of Exeter, Penryn, UK. [15] Institute of Evolutionary Ecology and Conservation Genomics, University of Ulm, Ulm, Germany. [16] Department of Zoology, University of Oxford, Oxford, UK. [17] Department of Community Ecology, Helmholtz Centre for Environmental Research–UFZ, Halle (Saale), Germany. [18] These authors jointly supervised this work: Roberto Salguero-Gómez, Tiffany M. Knight. ✉email: aldo.compagnoni@idiv.de

Climate change is altering the mean as well as the variance in temperature and precipitation worldwide[1]. These changes in climate are widely recognized as a prime global threat to biodiversity[2] because temperature and precipitation ultimately drive the demographic processes that determine the size and long-term viability of natural populations[3]. Hence, it is critical to evaluate which species are most responsive to climatic drivers, and in which biomes[4]. The urgency to understand the response of species to climate is particularly high for species that cannot buffer against the effects of climate change by migrating, such as sessile species. Among sessile species, numerous plants have short-distance dispersal, and cannot, therefore, shift their ranges fast enough to keep up with the current pace of climate change[5,6].

Assuming plant productivity is a proxy of population performance, we expect that precipitation, or its interaction with temperature, predicts plant population growth better than temperature alone. Most plant physiological processes, such as seed germination, tissue growth, floral induction, and seed set, are affected by water availability[7]. Accordingly, precipitation is a key driver of vegetation productivity worldwide[8]. Temperature can also influence these processes, but typically by modulating water availability[9], as plant growth occurs across a wide range of temperatures (namely between 5 and 40 °C[7,10]). The effect of temporal fluctuations on the growth rate of a population should be proportional to precipitation or temperature anomalies, where anomalies are deviations from mean values.

Precipitation and temperature anomalies are expected to have more pronounced effects in arid and cold biomes than in wet and temperate ones. While species should be adapted to their respective environment, extreme environments impose hard physiological limitations. In arid environments, plants experience water limitation more frequently[11]. Similarly, in cold biomes plants should more frequently experience temperatures that are too low to allow tissue growth[10,12]. Accordingly, as water availability decreases, precipitation becomes the main factor limiting plant physiological processes[13,14]. On the other hand, in cold biomes temperature anomalies can be disproportionately important. For example, the temperature has a positive effect on tree growth that increases in explanatory power with altitude[15,16]. Similarly, in the tundra temperature anomalies can dramatically change the length of the growing season[17]. However, because plant functional composition is filtered by biome[18], it is important to consider whether differences in the responses of plants across biomes might be due to the different composition of plant functional types (graminoids, herbs, ferns, woody species, and succulents) that occur in those biomes.

The life-history theory also provides expectations for how natural plant populations may respond to climate drivers. The key life-history trait defining plant life-history strategy is generation time, which describes how fast individuals in a population are substituted and is correlated with life expectancy[19]. The population growth of long-lived species should respond weakly to climatic anomalies compared to short-lived species. We expect this because the long-run population growth rate of long-lived species responds less strongly to increases in the temporal variation of survival, growth, and reproduction[20]. Here, we capitalize on the recent availability of large volumes of demographic data to quantitatively test how to plant population growth rate, $\lambda$, responds to temperature and precipitation anomalies. We expect (H$_1$) $\lambda$ to be more strongly associated with precipitation than temperature anomalies, because we expect water availability to having stronger physiological effects than temperature; (H$_2$) $\lambda$ of plants in water-limited biomes to be more responsive to precipitation anomalies; (H$_3$) $\lambda$ of plants in cold biomes to be more responsive to temperature anomalies; (H$_4$) species with greater

generation time to respond more weakly to temperature and precipitation anomalies. We show that the effect of precipitation is three times larger than that of temperature (H$_1$). Moreover, larger generation times are associated with weaker responses to climate (H$_4$). Both of these findings will inform ecological forecasts, and the result on generation time emphasizes the importance of this life-history trait to conservation assessments.

## Results

Our model selection provided little evidence for nonlinear responses to climate, and little evidence of interactions between climatic and non-climatic factors. A nonlinear model was selected in eight of the 38 populations for which we tested nonlinear relationships (Supplementary Figs. 3–5). We, therefore, considered a linear relationship for the remaining 154 populations; we present these linear relationships in the online repository that also contains the data and code related to this study[21]. Only two populations showed a substantial effect of the interaction between climate anomalies and covariates: our only population of *Astragalus cremnophylax* var. *cremnophylax*, and one of *Dicerandra frutescens* (Supplementary Data 1). These interactions increased the estimates of the climatic effect by 40 times (from 0.001 to 0.052) and decreased it by 16% (from −0.189 to −0.158), respectively.

**The overall effect of climate on plant population growth rate**. As predicted (H$_1$), the overall effect of precipitation anomalies on a log($\lambda$) was strong ($\beta = 0.031$, 95% CI: 0.007–0.054) relative to that of temperature ($\eta = -0.013$, 95% CI: −0.036 to 0.009) and their interaction ($\theta = -0.008$, 95% CI: −0.029 to 0.011), which were centered around zero. On average, a year with precipitation one standard deviation above the mean changed $\lambda$ by +3.3%.

**The effect of biome on the response of plants to climate**. The *meta*-regressions testing the response of plant populations to precipitation (H$_2$) and temperature (H$_3$) anomalies were both nonsignificant (Fig. 1). When testing the correlation between WAI and the response of plant populations to precipitation anomalies, only 90.5% of our bootstrap samples had slopes below zero ($\beta_{meta} = -3.83 \times 10^{-5}$, 95% CI: −9.47 × 10$^{-5}$, 1.99 × 10$^{-5}$). Similarly, we did not find evidence that the mean annual temperature (H$_3$) of the site predicted the response of plant populations to temperature anomalies (Fig. 1b; $\beta_{meta} = -1.42 \times 10^{-3}$, 95% CI: −6.62 × 10$^{-3}$, 1.00 × 10$^{-2}$).

**The effect of generation time on the response of plants to climate**. We found strong support for the effect of generation time (H$_4$) on the absolute response of plant populations to climate. As expected, the response of species to climate correlated negatively with generation time (Fig. 2). In these *meta*-regressions, 100% of simulated $\beta_{meta}$ values referring to the effect of precipitation ($\beta_{meta} = -0.54$, 95% CI: −0.63 to −0.44]), and temperature ($\beta_{meta} = -0.40$, 95% CI: −0.50 to −0.30]) were below zero.

**The effect of plant types on estimates of climate effects**. The effect of precipitation ($P < 0.01$), but not temperature ($P = 0.97$), changed based on organism type according to the ANOVA tests. Tukey's honestly significant difference test showed a significant difference in the effect of precipitation between herbaceous and graminoid species (Supplementary Tables 2 and 3, Supplementary Fig. 6). We, therefore, re-run separate tests of H$_2$, and H$_4$ excluding the precipitation effect sizes of graminoid species. We excluded graminoid species only because herbaceous species comprised 127 of our 162 populations so that excluding them

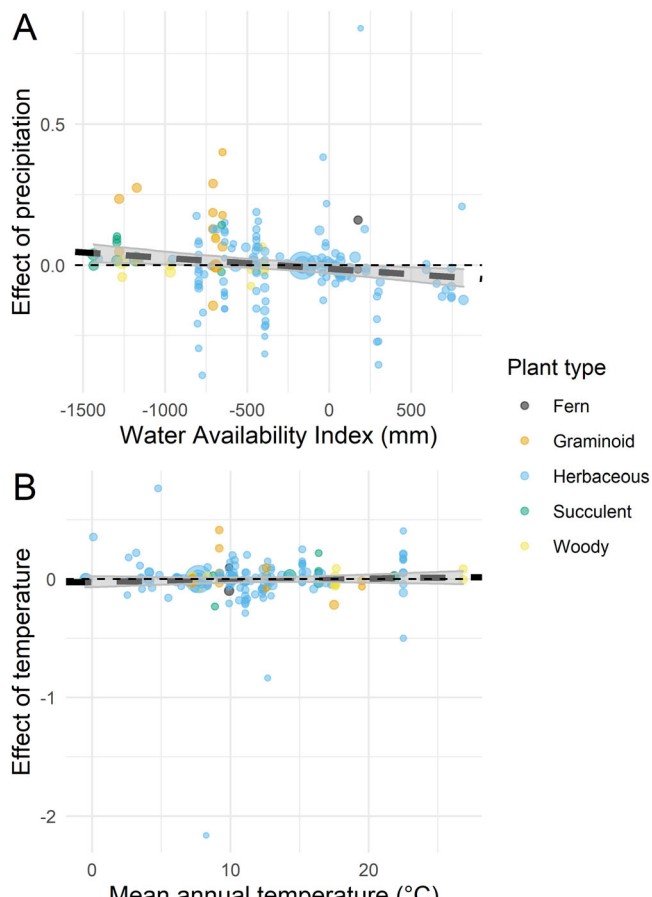

**Fig. 1 The effect of precipitation and temperature anomalies as a function of site mean aridity and temperature.** Effect of precipitation (**A**) and temperature (**B**) anomalies on the logged asymptotic population growth rate ($\lambda$) as a function of water availability index (**A**) and mean annual temperature (**B**). The y-axis represents the effect sizes of yearly anomalies in precipitation and temperature. The uncertainty of these effect sizes is shown by the size of circles, which are inversely proportional to the standard error (SE) of effect sizes (1/SE). The thick black lines show the mean prediction of the *meta*-regressions; these lines are dashed because these relationships are nonsignificant. The shaded areas represent the 95% confidence interval of 1000 bootstrapped linear regressions. The color of individual data points shows five separate plant types.

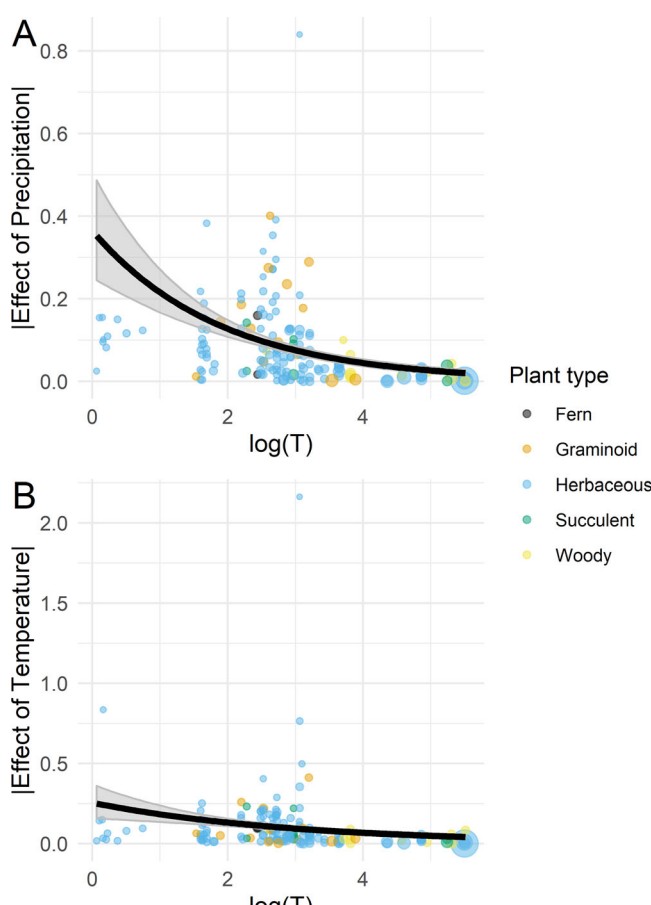

**Fig. 2 The absolute effect of precipitation and temperature anomalies as a function of logged generation time (*T*).** We show the effect sizes of precipitation and temperature anomalies as a function of log(*T*) (panels **A** and **B**, respectively). The uncertainty of these effect sizes is shown by the size of circles, which are inversely proportional to the standard error (SE) of effect sizes (1/SE). The thick black lines show the mean prediction of the *meta*-regressions. The shaded areas represent the 95% confidence interval of 1000 bootstrapped gamma regressions. The color of individual data points shows five separate plant types.

would not provide meaningful inferences. In these additional tests discarding graminoid data, $H_2$ was not supported, and $H_4$ was upheld. In $H_2$, the percentage of simulated $\beta_{meta}$ values lower than zero was 72%, well below the 90.4% of the full dataset (Supplementary Methods, Supplementary Fig. 7). On the other hand, $H_4$ was upheld, with 100% of $\beta_{meta}$ below zero (Supplementary Methods, Supplementary Fig. 8).

## Discussion

While quantifying population responses to climate drivers has a long history in plant ecology[22], there is an urgent need to synthesize our knowledge due to on-going climate change[4,23]. The availability of open-access data[24], a solid understanding of physiological ecology[25], and a mature evolutionary theory of life histories[26] provide opportunities to produce quantitative generalizations regarding plant population responses to climate. In our global synthesis, we found that ($H_1$) precipitation has a stronger effect on population growth rates than temperature and that ($H_4$) plant species with shorter generation time respond

more strongly to climate. These generalizations, especially the one on generation time, are relevant to conservation planning and evolutionary theory. However, because the available data is biased towards herbaceous perennials of temperate regions, our results might not be universal.

The large, positive effect of precipitation on a log($\lambda$) and the negative, smaller effects of temperature and its interaction with precipitation are consistent with the importance of water availability on plant population performance[25] and productivity[8]. The importance of precipitation as a driver of plant population growth implies highly uncertain ecological forecasts. Climate change projections involving precipitation are much more uncertain than those involving temperature[23]. Moreover, prediction uncertainty in climate projections is not expected to improve much in the coming decades[27]. As a result, accounting for this uncertainty will be a fundamental step when crafting ecological forecasts of plant populations (e.g., model uncertainty[28]).

To our knowledge, our results are the first to show that generation time is linked to population responses to climatic drivers across a large number of species. To our knowledge, the only other study to test for this hypothesis found a similar pattern for

three amphibian species[29]. We formulated our hypothesis linking generation time to population responses to climate because in a sample of long-lived plants and animals, Morris et al[20]. found that the long-run population growth rate responds little to increases in the variation of survival and reproduction. Our results are complementary to this seminal study, in that the low sensitivity to climate drivers we found in long-lived species should minimize the variation in yearly population growth rates. Such minimized variation in yearly population growth rates is linked to higher long-run population growth rates[30–32]. Hence, we demonstrated that it is possible to use plant traits to predict which species will be most sensitive to climate change[4]. Interestingly, generation time is a fundamental quantity in identifying extinction probability[33,34]. It is, therefore, a good news that this trait can also predict the climatic sensitivity of herbaceous plants.

The fact that responses to climate do not change based on biome suggests that plant populations are demographically adapted to cope with climate variation regardless of the average climate. In extreme environments, the stronger effect of climate on the variation of ecosystem processes such as productivity[14,35] or biomass accumulation[15,16] is not reflected in demographic patterns. It is, therefore, plausible that adaptations such as investment in survival[36] or dormancy[37] are sufficient to de-couple physiological processes from demographic patterns. Such de-coupling is crucial because if climate drove larger variance in population growth rates, this would decrease the chances of population persistence. However, because plants appear adapted to local climatic variation, these results do not mean that all biomes will be equally vulnerable to climatic change. Rather, vulnerability to climate change will likely depend on how changes in climate compare to pre-existing climatic variability[38].

The geographic and taxonomic bias of our dataset might amplify the relevance of precipitation anomalies, and it therefore may affect the generality of our findings. First, geographic bias potentially underemphasizes the role of temperature, because of our dataset under-samples extremely cold and hot biomes. For example, in cold biomes such as montane or boreal forests, the influence of temperature on growth is larger as the mean annual temperature decreases[15,16]. On the other hand, the interaction between precipitation and temperature may be larger in hot than in colder biomes[9]. Therefore, we might expect a strong interaction between precipitation and temperature anomalies where mean precipitation is low and mean temperature high. These conditions should occur, for example, in the subtropical desert or tropical savannas, but only a handful of our studies occur in these biomes (Supplementary Fig. 1). Similarly, the taxonomic bias in our data could also amplify the importance of precipitation anomalies. For example, our dataset contained only two trees and five shrubs. However, woody species have surprisingly effective adaptations to cope with water shortages[39], and they could therefore be susceptible only to extreme precipitation anomalies. Nevertheless, we note that inferences dominated by herbaceous perennials have high significance globally. At least 40% of terrestrial habitats are dominated by grasslands[40], herbaceous species comprise most of the biodiversity in temperate forests[41], and they have a critical role in the carbon cycle[42].

Our data on graminoids exemplify that the covariation between taxonomies and biomes complicates the interpretation of global comparative studies. In our results, the response of graminoids to precipitation anomalies is larger than other plant types, and this response drives the positive correlation between WAI and the effect of precipitation (Fig. 1a). Moderately arid climates favor grasses[43], which might have an inherent advantage in exploiting precipitation or at least precipitation pulses that increase the moisture of shallow soil horizons[11]. As a result, we cannot establish whether sensitivity to precipitation anomalies is

characteristic of graminoids, or, as we originally expected (H$_2$), of arid biomes. In future studies, disentangling the role of biomes and taxonomic bias on plant climate sensitivity will require study designs that stratify plant types across biomes.

The predictive ability of our results, which use as predictors of annual climatic anomalies calculated from gridded climatic data, could be improved in the future by mechanistic models that use increasingly more available microclimatic information[44]. Gridded climatic data are adequate to estimating climatic means registered by weather stations over long time periods, such as years[45]. However, the temperature experienced by plant tissues can sometimes be substantially different from the air temperature registered by weather stations[46,47]. We note, however, that this fact does not invalidate the use of gridded climatic data, because annual anomalies observed at the microclimatic and weather station level should be similar. For example, a previous study shows a tight linear relationship between air temperature and the microclimate at the leaf surface in alpine vegetation[47]. Nevertheless, microclimatic data will be required to test mechanistic models of climatic effects, such as those linked to thresholds. Examples of these thresholds are growing degree days[48] (Mcmaster 1997) or frost damage[49]. Similarly, microclimatic anomalies could help understand why different populations of the same species respond differently to comparable climatic anomalies[50].

Our findings on the link between short generation times and climatic sensitivity do not automatically translate into climate vulnerability. The observational nature of our data imposes to interpret our findings in light of two caveats. First, our data did not address several of the concurrent factors that contribute to the effects of climate on populations. These include factors such as density-dependence[3], trophic interactions[51], and anthropogenic drivers[52]. Second, our results are more relevant to changes in climatic variability than changes in climatic means. When predicting the effects of large changes in climatic means, our nonlinear results (Supplementary Figs. 3–5) show that extrapolation might not be warranted. Besides these two caveats, the conservation literature links short generation times to lower, rather than higher climate vulnerability as indicated by our results[53,54]. These studies reflect conservation assessments which posit that short generation time should be linked to lower extinction probability[33]. Short-generation time should also increase the probability of evolutionary rescue[55]. However, the advantages provided by short generation time might be over-ridden by the rapid rates of climate change expected. Thus, weighing the positive and negative effects of generation time will leverage our findings to improve the quality of climate change vulnerability assessments.

## Methods

**Demographic data**. To address our hypotheses, we used matrix population models (MPMs) or integral projection models (IPMs) from the COMPADRE Plant Matrix Database (v. 5.0.1[56]) and the PADRINO IPM Database[57], which we amended with a systematic literature search. First, we selected density-independent models from COMPADRE and PADRINO which described the transition of a population from 1 year to the next. Among these, we selected studies with at least six annual transition matrices, to balance the needs of adequate yearly temporal replicates and sufficient sample size for a quantitative synthesis. This yielded data from 48 species and 144 populations.

We then performed a systematic literature search for studies linking climate drivers to structured population models in the form of either MPMs or IPMs. We performed this search on ISI Web of Science for studies published between 1997 and 2017. We used a Boolean expression containing keywords related to plant form, structured demographic models, and environmental drivers (Supplementary Methods). We only considered studies linking macro-climatic drivers to natural populations (e.g., transplant experiments and studies focused on local climatic factors such as soil moisture, light due to treefall gaps, etc. were excluded). Finally, we used the same criteria used to filter studies in COMPARE and PARDINO, by selecting studies with at least six, density-independent, annual projection models.

This search brought two additional species, belonging to three additional populations, which we entered in the COMPADRE database.

One of the studies we excluded from the literature search because it contained density-dependent IPMs, also provided raw data with high temporal replication (14–32 years of sampling) for 12 species from 15 populations[58]. Therefore, we re-analyzed these freely available data to produce density-independent MPMs that were directly comparable to the other studies in our dataset (Supplementary Methods).

The resulting dataset consisted of 46 studies, 62 species, 162 populations, and a total of 3761 MPMs and 52 IPMs (Supplementary Data 1). The analyzed plant populations were tracked for a mean of 16 (median of 12) annual transitions. To our knowledge, this is the largest open-access dataset of long-term structured population projection models. However, this dataset is taxonomically and geographically biased. Specifically, among our 62 species, this dataset contains 54 herbaceous perennials (11 of which graminoids), and eight woody species: five shrubs, two trees, and one woody succulent (*Opuntia imbricata*). Moreover, almost all of these studies were conducted in North America and Europe (Supplementary Fig. 1), in temperate biomes that are cold, dry, or both cold and dry (Supplementary Fig. 1, inset). Our geographic and taxonomic bias reflects the rarity of long-term plant demographic data in general. This dearth of long-term demographic data is particularly evident in the tropics. The ForestGEO network[59] is an exception to this rule, but to date, no matrix population models or integral projection models using these data have been published.

We used the MPMs and IPMs in this dataset to calculate the response variable of our analyses: the yearly asymptotic population growth rate ($\lambda$). This measure is one of the most widely used summary statistics in population ecology[60], as it integrates the response of multiple interacting vital rates. Specifically, $\lambda$ reflects the population growth rate that a population would attain if its vital rates remained constant through time[61]. This metric therefore distills the effect of underlying vital rates on population dynamics, free of other confounding factors (e.g., transient dynamics arising from population structure[62]). We calculated $\lambda$ of each MPM or IPM with standard methods[61,63]. Because our MPMs and IPMs described the demography of a population transitioning from one year to the next, our $\lambda$ values were comparable in time units. Finally, we identified and categorized any non-climatic driver associated with these MPMs and IPMs. Data associated with 21 of our 62 species explicitly quantified a non-climatic driver (e.g., grazing, neighbor competition), for a total of 60 of our 162 populations. Of the datasets associated with these species, 19 included discrete drivers, and only three included a continuous driver.

**Climatic data**. To test the effect of temporal climatic variation on demography, we gathered global climatic data. We downloaded 1 km² gridded monthly values for maximum temperature, minimum temperature, and total precipitation between 1901 and 2016 from CHELSAcruts[64], which combines the CRU TS 4.01[65], and CHELSA[66] datasets. Gridded climatic data are especially suited to estimate annual climatic means[45]. These datasets include values from 1901 to 2016, which are necessary to cover the temporal extent of all 162 plant populations considered in our analysis. For our temperature analyses, we calculated the mean monthly temperature as the mean of the minimum and maximum monthly temperatures. We used monthly values to calculate the time series of mean annual temperature and total annual precipitation at each site. We then used this dataset to calculate our annual anomalies for each census year, defined as the 12 months preceding a population census. Our annual anomalies are standardized z-scores. For example, if $X$ is a vector of 40 yearly precipitation or temperature values, $E()$ calculates the mean, and $\sigma()$ calculates the standard deviation, we compute annual anomalies as $A = [X - E(X)]/\sigma(X)$. Therefore, an anomaly of one refers to a year where precipitation or temperature was one standard deviation above the 40-year mean. In other words, anomalies represent how infrequent annual climatic conditions are at a site. Specifically, if we assume that A values are normally distributed, values exceeding one and two should occur every 6 and 44 years, respectively. We used 40-year means because the minimum number of years suggested to calculate climate averages is 30[67].

Z-scores are commonly used in global studies on vegetation responses to climate[8,68], and they reflect the null hypothesis that species are adapted to the climatic variation at their respective sites. Across our populations, the standard deviations of annual precipitation and temperature anomalies change by 300% and 60%, respectively (Supplementary Fig. 2). Thus, a z-score of one refers to a precipitation anomaly of 50 or 160 mm and to a temperature anomaly of 0.5 or 0.8 °C. Our null hypothesis posits that species are adapted to these conditions, regardless of the absolute magnitude of the standard deviation in annual climatic anomalies. If this null hypothesis were true, each species would respond similarly to z-scores. Z-scores are more easily interpreted when calculated on normally distributed variables. We found our temperature and precipitation z-scores were highly skewed (skewness above 1) only in, respectively, 2 (for temperature) and three (for precipitation) of our 162 populations. We concluded that this degree of skewness should not bias our z-scores substantially.

To test how the response of plant populations to climate changes based on biome we used two proxies of water and temperature limitation. For each study population, we computed a proxy for water limitation, water availability index (WAI), and temperature limitation using mean annual temperature. To compute

these metrics, we downloaded data at 1 km² resolution for mean annual potential evapotranspiration, mean annual precipitation, and mean annual temperature referred to the 1970–2000 period. We obtained potential evapotranspiration data from the CGIAR-CSI consortium (http://www.cgiar-csi.org/). This dataset calculates potential evapotranspiration using the Hargreaves method[69]. We obtained mean annual precipitation and mean annual temperature from Worldclim[70]. Here, we used WorldClim rather than CHELSA climatic data because the CGIAR-CSI potential evapotranspiration data were computed from the former. We calculated the WAI values at each of our sites by subtracting mean annual potential evapotranspiration from the mean annual precipitation. Such proxy is a coarse measure of plant water availability that ignores information such as soil characteristics and plant rooting depth. However, WAI is useful to compare water availability among disparate environments, so that it is often employed in global analyses[68,71]. As our proxy of temperature limitation, we use mean annual temperature. While growing degree days would be a more mechanistic measure of temperature limitation[48], this requires daily weather data. However, we could not find a global, downscaled, daily gridded weather dataset to calculate this metric.

**The overall effect of climate on plant population growth rate**. To test $H_1$, we estimated the overall effect sizes of responses to anomalies in temperature, precipitation, and their interaction with a linear mixed-effect model.

$$\log(\lambda) = \alpha + \beta P + \eta T + \theta PxT + \varepsilon \qquad (1)$$

where $\log(\lambda)$ is the log of the asymptotic population growth rate of plant population $P$ is precipitation, $T$ is temperature. We included random population effects on the intercept and the slopes to account for the nonindependence of measurements within populations. We then compared the mean absolute effect size of precipitation, temperature, and their interaction. This final model did not include a quadratic term of temperature and precipitation because these additional terms led to convergence issues. This likely occurred because single data sets did not include enough years of data.

**Population-level effect of climate on plant population growth rates**. To test our remaining three hypotheses, we carried out *meta*-regressions where the response variable was the slope (henceforth "effect size") of climatic anomalies on the population growth rate for each of our populations. Before carrying out our *meta*-regression, we first estimated the effect size of our two climatic anomalies on the population growth rate of each population separately. We initially fit population-level and *meta*-regression simultaneously, in a hierarchical Bayesian framework. However, these Bayesian models shrunk the uncertainty of the noisiest population–level relationships, resulting in unrealistically strong *meta*-regressions. We, therefore, chose to fit population models separately, resulting in more conservative results.

For each population, we fit multiple regressions with an autoregressive error term, and we evaluated the potential for nonlinear effects in the datasets longer than 14 years. We fit multiple regressions because temperature and precipitation anomalies were negatively correlated, so that fitting separate models for temperature and precipitation would yield biased results[72]. We fit an autoregressive error term because density dependence and autocorrelated climate anomalies can produce autocorrelated plant population growth rates. The form of our baseline model was

$$\log(\lambda)_y = \alpha + \beta_p P_y + \beta_t T_y + \varepsilon_y \qquad (2)$$

$$\varepsilon_y = \rho \varepsilon_{y-1} + \eta_y \qquad (3)$$

The model in Eq. 2 is a linear regression relating each $\log(\lambda)$ data point observed in year $y$, to the corresponding precipitation ($P$) and temperature ($T$) anomalies observed in year $y$, via the intercept $\alpha$, the effect sizes, $\beta$, and an error term, $\varepsilon_y$, which depends on white noise, $\eta_y$, and on the correlation with the error term of the previous year, $\rho$. When multiple spatial replicates per each population were available each year, we estimated the $\rho$ autocorrelation value separately for each replicate. This happened in the few cases when a study contained contiguous populations, with no ecologically meaningful (e.g., habitat) differences.

We compared the baseline model in Eqs. 2 and 3 to models including a quadratic climatic effect and non-climatic covariates. We estimated quadratic climatic effects only for time series longer than 14 years. We choose this threshold because when using a model selection approach to select a quadratic or linear regression model, the recommended minimum sample size is between 8 and 25 data points[73]. We fit models including a quadratic effect of temperature, precipitation, or both (Supplementary Table 1).

Finally, we also tested whether non-climatic covariates could bias the effects of climate on $\log(\lambda)$ estimated in our analysis. Such bias, either upwards or downwards, could result in the case non-climatic co-variates interacted with climate. For example, harvest can have multiplicative, rather than additive effects on the climate responses of forest understory herbs[74]. We tested for an interaction between a covariate and climate anomaly in 17 of the 21 studies that included a non-climatic covariate. In the remaining three studies, discrete covariates corresponded with the single populations. Because Eqs. 2 and 3 is fit on separate populations, it implicitly accounted for these covariates. For the 17 studies above,

we fit a linear effect of the non-climatic covariate and its interaction with one of the two linear climatic anomalies. Thus, including the linear model in Eqs. 2 and 3, the nonlinear models, and the covariate interaction models, we tested up to six alternative models for each one of our populations (Supplementary Table 1). We selected the best model according to the Akaike Information Criterion corrected for small sample sizes (AICc[75]). We carried out these and subsequent analyses in R version 3.6.1[76].

In the populations for which AICc selected one of the model alternatives to the baseline in Eqs. 2 and 3, we calculated the effect size of climate by adding the effect of the new terms to the linear climatic terms. For example, when a quadratic precipitation model was selected, we calculated the effect size of precipitation as $\beta = \beta_p + \beta_{p2}$. For models including an interaction between temperature and a non-climatic covariate, we evaluated the effect of the interaction at the mean value of the covariate. Therefore, we calculated the effect size as $\beta = \beta_t + \beta_x E(C_i)$ for continuous covariates. For categorical variables, we calculated the effect size as $\beta_p + \beta_x 0.5$: that is, we calculated the mean effect size between the two categories. We quantified the standard error of the resulting effect sizes by adding the standard errors of the two terms.

**The effect of biome on the response of plants to climate**. We used a simulation procedure to run two *meta*-regressions to test for the correlation between the effect size of climate drivers on $\lambda$, and our measures of water or temperature limitation. These *meta*-regressions accounted for the uncertainty, measured as the standard error, in the effect sizes of climate drivers. We represented the effect of biome using a proxy of water (WAI) and temperature (mean annual temperature) limitation. For each of our 162 populations, the response data of this analysis were the effect sizes ($\beta_p$ or $\beta_t$ values) estimated by Eqs. 2 and 3 or their modifications in case a quadratic or non-climatic covariate model were selected. In these *meta*-regressions, the weight of each effect size was inversely proportional to its standard error. To test H₂ and H₃ on how water and temperature limitation should affect the response of populations to climate, we used linear *meta*-regressions. These two hypotheses tested both the sign and magnitude of the effect of climate. Therefore, we used the effect sizes as a response variable which could take negative or positive values. As predictors, we used population-specific WAI (H₂, only for effect sizes quantifying the effect of precipitation), and mean annual temperature (H₃, only for effect sizes quantifying the effect of temperature). The null hypothesis of these *meta*-regressions is that plant species are adapted to the climatic variation at their respective sites. Such an adaptation implies that a precipitation $z$-score of one should produce effects on log($\lambda$) of similar magnitude and sign across different climates. This should happen across average climatic values that are connected to substantially different absolute climatic anomalies (Supplementary Fig. 2). On the other hand, our hypotheses posit that at low WAI and MAT values, species are more responsive to $z$-scores than expected under the null hypothesis.

We performed these two *meta*-regressions by exploiting the standard error of each effect size. We simulated 1000 separate datasets where each effect size was independently drawn from a normal distribution whose mean was the estimated $\beta$ value, and the standard deviation was the standard error of this $\beta$. These simulated datasets accounted for the uncertainty in the $\beta$ values. We fit 1000 linear models, extracting for each its slope, $\beta_{meta}$. Each one of these slopes had in turn an uncertainty, quantified by its standard error, $\sigma_{meta}$. For each $\beta_{meta}$, we then drew 1000 values from a normal distribution with mean $\beta_{meta}$ and standard deviation $\sigma_{meta}$. We used the resulting $1 \times 10^6$ values to estimate the confidence intervals of $\beta_{meta}$. This procedure assumes that the distribution of $\beta_{meta}$ values is normally distributed. We performed one-tailed hypothesis tests, considering meta-regression slopes significant when over 95% of simulated values were below zero.

**The effect of generation time on the response of plants to climate**. To test H₄ on how the generation time of a species should mediate its responses to climate, we used a gamma *meta*-regression. We fitted gamma *meta*-regressions because our response variables were the absolute effect sizes of precipitation and temperature anomalies, |$\beta$|, which are bounded between 0 and infinity. To test H₄, we therefore fit gamma *meta*-regressions with a log link, using |$\beta$| values as response variable and generation time ($T$) as predictor. We calculated $T$ directly from the MPMs and IPMs (Supplementary Methods). We log-transformed $T$ to improve model fit. We carried out these meta-regressions using the same simulation procedure described for testing H₂ and H₃. We also carried out one-tailed hypothesis tests, by verifying whether 95% of $\beta_{meta}$ values were below zero.

**The effect of plant types on estimates of climate effects**. We verified whether certain plant types could bias our results by subdividing our species as graminoids, herbaceous perennials, ferns, woody species (shrubs and trees), and succulents. We ran ANOVA tests to verify whether the effect sizes of precipitation and temperature anomalies differed between plant types. We then tested for significant differences in pairwise contrasts between plants types by running Tukey's honestly significant difference tests. We carried out these tests on the average effects of climate, without accounting for differences in parameter uncertainty. If Tukey's test identified significant differences among plant types, we ran additional tests of H₂–H₄ excluding the plant type, or plant types, whose response to climate differed.

**Reporting summary**. Further information on research design is available in the Nature Research Reporting Summary linked to this article.

## Data availability
Most of the demographic data used in this paper are open-access and available in the COMPADRE Plant Matrix Database (v. 5.0.1; https://compadre-db.org/Data/Compadre). Additional data come from the PADRINO Database (beta version; https://github.com/levisc8/rpadrino). A list of the studies and species used here is available in Supplementary Data 1. The CHELSAcruts dataset is available at 10.16904/envidat.159. The formatted dataset, and associated metadata, to reproduce the analyses of this study are archived on Github at https://doi.org/10.5281/zenodo.4516446.

## Code availability
The code to reproduce the results of this study is stored on Github at https://doi.org/10.5281/zenodo.4516446.

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

## Acknowledgements

This article is the result of working group sAPROPOS (Analyses of PROjectionS of POpulations) supported by sDiv, the Synthesis Centre of the German Centre for Integrative Biodiversity Research (iDiv) Halle-Jena-Leipzig (funded by the German Research Foundation, FZT 118—202548816), led by R.S.-G. and T.M.K. T.M.K., A.C., and S.L. were supported by the Alexander von Humboldt foundation; R.S.-G. was supported by NERC IRF (NE/M018458/1); J.C.-C., R.S.-G., and O.R.J. were also supported by an NSF Advances in Biological Informatics grant (DBI-1661342000), N.R. was funded by a research grant from Deutsche Forschungsgemeinschaft DFG (RU 1536/3-1). We acknowledge the efforts of the Max Planck Institute for Demographic Research in curating and making the COMPADRE Plant Matrix Database open-access, as well as the numerous authors who have kindly shared their demographic data and population models.

## Author contributions

T.M.K. and R.S.-G. conceived the study; T.M.K., A.C., R.S.-G., S.L., and M.P. designed the research; A.C., R.S.-G., S.L., and T.M.K. performed the research; A.C., R.S.-G., S.L., G.R.

analyzed the data; A.C., R.S.-G., T.M.K., and S.L. wrote the first draft of the article with contributions by D.C. and S.H.; A.C., S.L., D.Z.C., S.H., M.P., G.R., J.H.B., J.C.C., N.R., G.K., J.M.B., C.R.A., O.R.J., R.S.G., and T.M.K. contributed to the final version of the paper.

## Funding

## Competing interests
The authors declare no competing interests.
