## [Peer Review File · Nature Communications]

REVIEWER COMMENTS

Reviewer #1 (Remarks to the Author):

This submission is a synthesis of plant population behaviour with the intent to identify the role of climate variation in mediating plant population dynamics. They have assembled an impressive data collection on population dynamics as well as gridded site climate information, and use these data to analyse the responses to variability in annual temperature and precipitation among years. The authors contrast generation time and reproductive frequency as possible ways that population dynamics may respond to climate. The main message is that species with short generation time have demonstrably stronger absolute responses to climate anomalies than those with longer generation times, as embedded in the title and the Abstract. This finding has interesting implications for the adaptability of populations to climate change, as well as for reproductive trait evolution, although these aspects aren't discussed much.

The authors use comparative terms without letting the reader know what they are comparing to. As I read the title, I asked myself what was meant by 'stronger demographic response'. Stronger than what? I'm not clear on what they seek to compare their short-generation species with. Longer-generation, non-woody plants (because that is what their data comprise, see following)? Meadow plants (as that seems to be primarily what is involved in the study)? This aspect could be improved.

The first 4 paragraphs read with all sorts of generalities regarding 'plants'. The final sentence L. 490 reads "providing the first generalizations on the climate sensitivity of plant populations". I find many of these need to be qualified to 'herbaceous plants' or 'herbaceous plant population behaviour'. The database involves mostly herbaceous species and graminoids – to knowledge, there is only one woody species out of 62 species involved, and perhaps some sub-woody plants like *Solidago*. If the authors truly seek to generalise plant behaviour with respect to climate on this basis, whilst 45-50% of higher plants on earth are woody, then this limitation to the analysis should be clear to the reader. I don't see this ('herbaceous') as a qualifier anywhere, yet it is a very key one, since it would seem to bias against longer-lived behaviour and toward the shortening of generation times that they eventually found. It's also not clear if plants are all perennials or if there are annual plants, but I would have expected different implications to annual population dynamics for these groups. Similarly, there are 11 graminoid species, and the vegetative reproductive behaviour of graminoids ('tillering') might have biased results. Did the authors test for these things? There isn't much description of what sort of plants were analysed in Table S1, and this reviewer could accept that the type of plant can be ignored in the analysis, if that is proven by the analyses. Instead, consideration of the type of plant doesn't seem to have been addressed in the analysis.

This certainly is not a broad cross-section of plant types involved in the study, nor is it geographically broad. I see that there is only one site located at a latitude less than 20 degrees north, and the rest of the world seems not included. Perhaps these data are rarer, or non-existent for the Southern Hemisphere, but it appears biased. Both the span of plant types in the study and their geographical extent seem to suggest a very limited universe of inference that can be made from the study.

Detailed comments

The dataset is relatively small for what one might expect of such a data synthesis: dataset consisted of 62 species and 165 populations.

L. 82: As a plant physiologist, I can't find support for the statement that 'precipitation predicts plant population growth best, and secondarily modified by temperature'. Perhaps for herbaceous plants this is the case, when one considers water availability (NOT precipitation, as they stated) and the processes they are considering, like germination and recruitment. The point has been

made earlier the authors need to be clearer about their universe of inference – the statement is not about all plants, but rather forbs and graminoids.

L. 92: Water availability is NOT the difference between AET and precipitation. Water availability to plants also depends on their rooting zone and depth, as well as soil texture, since clays hold water unavailable more than sands, for instance. I accept the statement as a very broad simplification, but I think the authors should demonstrate that their characterisation is very coarse.

I didn't get a good sense of what comprised a climate anomaly (A in Eq. 3). Yes, that was defined on L. 90, and how computed described L. 178-184. They use gridded monthly values (L. 171), which by their nature would be dampened in terms of anomalies (because they are fitted and smoothed to obtain grids) compared with the real climate data. I suspect that sites in their analysis could also have World Met Organisation-standard data actually measured nearby. Were there any comparisons of actual versus gridded climate anomalies for such instances that would support using gridded data as a reasonable representation? Also, why wasn't mean annual precipitation used in Fig. 1A, which would have been more consistent with mean annual temperature? If precipitation is transformed to the water availability index, then shouldn't temperature be transformed, e.g. cumulative degree days.

Reviewer #2 (Remarks to the Author):

The main strength of the paper is that provides important information about the effects of variation in temperature and precipitation on projected population growth rates in plants by examining currently available information from long-term demographic studies of plants.

I do, however, think that the paper needs to, or can, be improved in several respects.

First, I think that the reasoning regarding the expected relationship between biomes and effects of climatic anomalies – hypotheses 2 and 3 - is a bit simplistic. As argued by the authors, it is reasonable to assume that effects of drought or temperature anomalies will be more severe when conditions are already dry or temperatures high. However, while this is true for a population with a given genetic composition, we might also expect that in arid or high-temperature areas, species which are sensitive to drought or high temperatures might already have been filtered out, or populations might have adapted locally to resist extreme conditions.

Second, I think that the logic behind the hypothesized relationship between parity and sensitivity to extreme weather is poorly explained and hard to understand. In fact, the authors admit in the discussion that their hypothesis might be wrong. Given this, and that relationships with parity and longevity are examined in separate statistical models although there are good reasons to expect that they are strongly correlated, I think that hypothesis 5 should be dropped.

Thirdly, I think that the part about temporal replication should can be omitted from the paper. This is of two reasons. The first is that I think that this question is conceptually poorly connected to the rest of the study. The second, and most important, reason is that the analyses carried out by the authors does not tell us anything about how temporal replication affects the ability to correctly assess the effects of anomalies in temperature or precipitation on population growth rate, only about how much of the proportion of climatic variation that is sampled. If relationships between climatic factors and population growth rates are linear, which is assumed in all statistical models fitted by the authors, then it is not the range of climatic variation sampled that determines the ability to correctly assess the effects. It is instead the magnitude of temperature effects vs. the effects of other environmental drivers and sampling errors. If relationships can be correctly estimated for a smaller range of climate conditions, then only climate data are necessary to extrapolate effects of more extreme weather conditions as long as relationships are linear.

This brings me to my fourth concern; the lack of tests for non-linear relationships and the lack of discussion of the likelihood that effects of climate might be non-linear and that effects of extreme weather conditions cannot always be predicted from relationships under more restricted range of

variation. I realize that for many of the species, the temporal replication is too poor to allow for such a test. However, for several species in your data set the number of years is clearly sufficient to test for the presence of non-linear relationships.

Fifth, a large number of recent studies that have demonstrated that the climate experienced locally by, for example a plant population, might differ dramatically from the regional climate, and that several features of the physical and biotic environment might buffer variation in the regional climate. Against this background it would be interesting to discuss the limitation of studies, such as this, that are based on gridded temperature data.

Sixth, I think that the authors need to better explain why they consider z-values as biologically relevant measures of climatic variation in this context, and in what respect this measure might be problematic. For example, I imagine that for some populations, yearly values of precipitation and temperature might be quite skewed.

Seventh, I think that the authors need to be much more careful when interpreting their results in terms of predicting responses to climate change (e.g. lines 447-8) as increases in standardized anomalies are not necessarily equivalent to predicted changes in climate. For example, mean conditions might often be more important to long-term population growth rate than variance in conditions.

More specific comments:

Lines 105-7: It seems hard to believe that the population growth rate of long-lived species (with higher survival elasticity) should be less sensitive to changes in survival than short-lived species (with higher reproduction elasticity). If you maintain that this statement is correct, then I think it needs further explanation.

Lines 245-62: It is unclear to me on what sort of logic you base these calculations. What kind of problem does effects of other "confounding" environmental factors pose, and how do you "correct" for it by carrying out the calculations you did? Also, what are the implications of that you accounted for the very tiny proportion of effects of non-climatic drivers that were measured and reported in the studies, but neglect the vast majority of effects that were not measured. Lastly, what did you gain from your adjustments? How much did adjusted values differ from raw means?

Lines 348-55 and corresponding parts of the discussion: You use much space to report and discuss this non-significant result, and you also start by describing the effect and only 4 lines lower down tell that it was not significant.

Lines 442-5: This reasoning is very simplistic and seems to neglect the role of trade-offs in life history evolution.

Lines 480-491: This discussion of the pros and cons of demographic approaches and Species Distribution Models is quite superficial and does not add much to the paper.

The discussion lacks references to some recent papers exploring effects of climatic factors on plant population growth rate.

Table S1 lists one 5-year study although the text says that only studies with 6 or more years were included. It also lists climate as a non-climatic factor (*Crypthanta flava*).

Reviewer #3 (Remarks to the Author):

Dear Editor,

The manuscript "Short-lived plants have stronger demographic responses to climate" is a synthesis of empirical evidence on plant population responses to climatic changes in temperate biomes. The major strength of the study lies in the analysis of the longest time series of demographic observations available to date in plant demography databases. It is sobering that in the areas of the world the most intensely researched the authors found only 46 studies (62 species and 165 populations) that have monitored populations for at least seven years. Even this monitoring effort, as the authors demonstrate it, is not really enough to fully understand plant population responses to the threats of climate change. While responses sought at the biome-level still prove noisy, we can tap on species' evolutionary history to reveal global signals of climate effects on population

performance. This is a very strong message that should incentivise longer term demography research, especially in biomes where such data is barely available.

I have three, or perhaps four major concerns about the approaches taken:

1) Hypothesis 2 and Hypothesis 3 refer to effects of temperature and precipitation anomalies on species of cold and arid biomes respectively. It is still not clear how the response variable in the meta-regression is independent of the predictor variables, it would be great if this was explained upfront in the text. Further, I was expecting an interaction term between temperature and precipitation in meta-regression, since the authors suggest already in the introduction that such an interaction should be expected, and in H1 even have this term included. I am also puzzled by Eq. 2 and 3, where I was expecting an autocorrelation term among the explanatory variables, since it is well known in demography that yearly growth rates (any temporal measurements) are likely autocorrelated. Overlooking this term in these two types of model could be at the root of weak climate signals obtained.

2) Yearly autocorrelation has also been overlooked when calculating the response variable, lambda. It is well known in demography that population growth rate can differ when calculated with or without accounting for the effect of (mostly) climate autocorrelation on population performance.

3) While the population growth rate is indeed the most widely used population measure, it is well known in demography that populations are able to buffer both the temporal and the spatial variation in its vital rates. While we surely wish first and foremost to obtain signals from this integrated population measure, existing evidence makes me suspect that more sensitive responses, of the vital rates, have remained hindered in this dataset. I think there would be purpose for a couple of more detailed analyses at least for H2 and H3, where obviously there is need for more in depth explorations.

4) Plant responses even to climate anomalies are not likely to be linear. I appreciate that the authors at least tried to include a quadratic term when testing for H1, and that in Eq 2 and 3 a quadratic term would have been again problematic for most populations. However, would it be possible to look for any evidence in the data for such a non-linear relationship, perhaps in the populations studied over the longest term?

Further minor notes:

L85: It is still a puzzle to read this generalisation in this context. Plants do have temperature and drought tolerance curves, and in the context it is sensitive to generalise about how temperature modulates water availability equally across biomes and life histories.

L108: It would be welcome at this point to learn more about the findings in Demetrius 1978.

L172: Is year 1901 a mistake?

L174: I don't agree with this way of calculating the mean monthly temperature, it should be correctly calculated from daily values.

L410: What is the limiting factor then?

L475: In Csergo et al. 2017 there is also a great deal of agreement on that the population growth rate is often not the perfect measure of plant responses to climate or habitat suitability. This sentence is best softened, since similar types of disagreement may be envisaged with respect to population responses to climate (or the way we calculate climate variables). Having said these, I agree that generalisations on climate sensitivity directly are highly needed and open up new avenues in comparative demography research.

Below, we provide line numbers for passages that refer to two versions of the article: **BOLD** line numbers (e.g. **L. 161-162**) refer to the word document containing all markup tracked changes, regular font, underlined line numbers (e.g. L. 161-162) refer to the document converted into PDF form. Furthermore, we provide our responses point by point in the format “**RX**”, where X is the response number.

Reviewer #1 (Remarks to the Author):

This submission is a synthesis of plant population behaviour with the intent to identify the role of climate variation in mediating plant population dynamics. They have assembled an impressive data collection on population dynamics as well as gridded site climate information, and use these data to analyse the responses to variability in annual temperature and precipitation among years. The authors contrast generation time and reproductive frequency as possible ways that population dynamics may respond to climate. The main message is that species with short generation time have demonstrably stronger absolute responses to climate anomalies than those with longer generation times, as embedded in the title and the Abstract. This finding has interesting implications for the adaptability of populations to climate change, as well as for reproductive trait evolution, although these aspects aren't discussed much.

R1> We thank the reviewer, which pointed out we did not address the potential for evolutionary rescue in our discussion. We added a consideration on this topic in the last paragraph of the article (**L. 630-634**; L. 504-508):

These studies reflect conservation assessments which posit that short generation time should be linked to lower extinction probability (Mace et al. 2008). Short-generation time should also increase the probability of evolutionary rescue (Lynch 1993). However, the advantages provided by short generation time might be overridden by the rapid rates of climate change expected.

The authors use comparative terms without letting the reader know what they are comparing to. As I read the title, I asked myself what was meant by 'stronger demographic response'. Stronger than what? I'm not clear on what they seek to compare their short-generation species with. Longer-generation, non-woody plants (because that is what their data comprise, see following)? Meadow plants (as that seems to be primarily what is involved in the study)? This aspect could be improved.

R2> We have addressed the confusion regarding the use of the word "stronger" in the title, "***Perennial plants with short generation time have stronger responses to climate anomalies than those with longer generation time***".

We do not mention the term "herbaceous" in the title, because our dataset contains several perennial plants types, including woody species, succulents, ferns, and graminoids. We discuss how we addressed the issues linked to our taxonomic bias in **R3**, immediately below.

The first 4 paragraphs read with all sorts of generalities regarding 'plants'. The final sentence L. 490 reads "providing the first generalizations on the climate sensitivity of plant populations". I find many of these need to be qualified to 'herbaceous plants' or 'herbaceous plant population behaviour'. The database involves mostly herbaceous species and graminoids – to knowledge, there is only one woody species out of 62 species involved, and perhaps some sub-woody plants like Solidago. If the authors truly seek to generalise plant behaviour with respect to climate on this basis, whilst 45-50% of higher plants on earth are woody, then this limitation to the analysis should be clear to the reader. I don't see this ('herbaceous') as a qualifier anywhere, yet it is a very key one, since it would seem to bias against longer-lived behaviour and toward the shortening of generation times that they eventually found. It's also not clear if plants are all perennials or if there are annual plants, but I would have expected different implications to annual population dynamics for these groups. Similarly, there are 11 graminoid species, and the vegetative reproductive behaviour of graminoids ('tillering') might have biased results. Did the authors test for these things? There isn't much description of what sort of plants were analysed in Table S1, and this reviewer could accept that the type of plant can be ignored in the analysis, if that is proven by the analyses. Instead, consideration of the type of plant doesn't seem to have been addressed in the analysis.

R3> We agree with the reviewer that this aspect of our data warranted a systematic analysis. To address this comment, we updated our analyses, figures, tables, and manuscript text. We summarise these actions in six points:

1. In Table S1, we classified our species in plant types (trees, shrubs, succulents, ferns, graminoids, herbaceous perennials), and as in Dicotyledons/Monocotyledons.
2. We explicitly quantify our bias in plant types in the methods (L. 166-168, L. 141-143) and highlight it in the abstract (L. 60-61, L. 59-60):

L. 166-168, L. 141-143: among our 62 species, this dataset contains 50 herbaceous perennials (11 of which graminoids), five shrubs, four succulents, two trees, and one fern.

L. 60-61, L. 59-60: Here, we synthesized time series of structured population models from 162 populations from 62 plants, mostly herbaceous species from temperate biomes.

3. In the discussion, we have pointed out that the low number of woody species (seven) in our dataset decreases the confidence when applying our results to woody species (L. 593-597, L. 467-471):

Similarly, our taxonomic bias could also amplify the importance of precipitation anomalies. For example, our dataset contains only two trees and five shrubs. However, woody species have surprisingly effective adaptations to cope with water shortages (Dietrich et al. 2018), and they could therefore be susceptible only to extreme precipitation anomalies

4. We have added a new paragraph in the discussion to address the implications of the taxonomic and geographic biases in our dataset (L. 582-601, L. 456-475):

The geographic and taxonomic bias of our dataset might amplify the relevance of precipitation anomalies, and it therefore might affect the generality of our findings. First, our geographic bias potentially underemphasizes the role of temperature, because our dataset under-samples extremely cold and hot biomes. For example, in cold biomes such as montane or boreal forests, the influence of temperature on growth is larger as mean annual temperature decreases (Galván et al. 2014; Primicia et al. 2015). On the other hand, in hot biomes, the interaction between precipitation and temperature should become larger than in colder biomes (Aparecido et al. 2020). Therefore, we might expect a strong interaction between precipitation and temperature anomalies where mean precipitation is low and mean temperature high. These conditions should occur, for example, in subtropical desert or tropical savannas; however only a handful of available demographic studies occur in these biomes (Fig. S1). Similarly, our taxonomic bias could also amplify the importance of precipitation anomalies. For example, our dataset contained only two trees and four shrubs. However, woody species have surprisingly effective adaptations to cope with water shortages (Dietrich et al. 2018), and they could therefore be susceptible only to extreme precipitation anomalies. Nevertheless, we note that inferences dominated by herbaceous perennials have high global significance. At least 40% of terrestrial habitats are dominated by grasslands (Gibson 2009), herbaceous species comprise most of the biodiversity in temperate forests (Gilliman 2007), and they have a critical role in the carbon cycle (Scurlock and Hall 1998).

5. We changed Figure 1 and 2 in the main text, and the associated figures in the appendix (see point 6) to show which plant type each circle belongs to.
6. We tested differences between the estimates of precipitation and temperature effects across five plant types: woody species (shrubs and trees), succulents, ferns, graminoids, and other herbaceous perennials (in Methods: L. 403-412, L. 344-353; in Results: L. 486-497, L. 389-400; Appendix S5: Fig. S6; Table S3-4). Because we found statistically significant differences between the effects of precipitation in herbaceous and graminoid species (Appendix S5: Fig. S6; Table S3-4), we re-run the meta-regressions involving precipitation effects excluding effect sizes from graminoid species (Fig. S7-8).

In the methods (L. 403-412, L. 344-353):

The effect of plant types on estimates of climate effects

We verified whether certain plant types could bias our results by subdividing our species in graminoids, herbaceous perennials, ferns, woody species, and succulents. We ran ANOVAs to verify whether the effect sizes of precipitation and temperature anomalies differed between species types. Then, we ran Tukey's

honestly significant difference tests on the differences among plant types. Note that we carried out these tests on the average effects of climate, without accounting for differences in parameter uncertainty. If Tukey's test identified significant differences among plant types, we ran additional tests of H_2 , H_3 , and H_4 excluding the plant type or plant types whose response to climate differed.

In the results (L. 486-497, L. 389-400):

The effect of plant types on estimates of climate effects

The effect of precipitation ($P < 0.01$), but not temperature ($P=0.97$), changed based on organism type according to the ANOVA tests. Tukey's honestly significant difference test showed a significant difference in the effect of precipitation between herbaceous and graminoid species (Table S3-4, Fig. S6). We therefore re-run separate tests of H_2 , and H_4 excluding the effect sizes of graminoid species. We excluded graminoid species only, because herbaceous species comprised 127 of our 162 populations, so that excluding them would not provide meaningful inferences. In these additional tests discarding graminoid data, H_2 was not supported, and H_4 was upheld. In H_2 , the percentage of simulated β meta values lower than zero was 72%, well below the 90.4% of the full dataset (Appendix S5, Figure S7). On the other hand, H_4 was upheld, with 100% of β meta below zero (Appendix S5, Figure S8).

This certainly is not a broad cross-section of plant types involved in the study, nor is it geographically broad. I see that there is only one site located at a latitude less than 20 degrees north, and the rest of the world seems not included. Perhaps these data are rarer, or non-existent for the Southern Hemisphere, but it appears biased. Both the span of plant types in the study and their geographical extent seem to suggest a very limited universe of inference that can be made from the study.

R4> We agree with the reviewer's concerns regarding our geographic and taxonomic bias. We have addressed this comment in two ways:

1. Along with the change in the title of the article (**R2**), we also qualified the geographic scope of our study in the abstract (L. 60-61, L. 59-60; **R3** point 2), and the discussion (L. 582-601, L. 456-475; **R3** point 4).
2. In the methods, we now state that this is the largest open-access dataset of its kind (L. 164-165, L. 139-140):

To our knowledge, this is the largest open-access dataset of long-term structured population projection models.

3. Furthermore, we now acknowledge the rarity, and therefore the value of long-term demographic time series (L. 170-174, L. 145-149):

Our geographic and taxonomic bias reflects the rarity of long-term plant demographic data in general. This dearth of long-term demographic data is particularly true in the tropics. The ForestGEO network (Anderson-Teixeira et al. 2015) is an exception to this rule, but to date, no matrix population models or integral projection models using these data have been published.

Detailed comments

The dataset is relatively small for what one might expect of such a data synthesis: dataset consisted of 62 species and 165 populations.

R5> As noted above, this is the largest open access dataset of its kind, and it is up to date. At the onset of our study, we performed a systematic literature review to identify and digitize the most recent demographic studies linking plant demography to climatic drivers. We entered these studies in COMPADRE or PADRINO databases. The resulting versions of the COMPADRE and PADRINO databases that we use in this contain 760 and 76 species, respectively. However, filtering for the studies that contain at least six years of data produces the dataset that we present in this article.

We know of two, larger demographic databases that have been used in recent synthesis studies, but these datasets do not conform to our requirements. The first is made of long-term studies, but it is mostly proprietary (Shefferson *et al.* 2018). However, in our studies we decided to use open-access data only, as the COMPADRE and PADRINO are open-access databases. The second database was used in a recent article by Morris *et al.* (2020). This second database, however, does not examine climate using a regression design, and it therefore has smaller temporal replication requirements.

L. 82: As a plant physiologist, I can't find support for the statement that 'precipitation predicts plant population growth best, and secondarily modified by temperature'. Perhaps for herbaceous plants this is the case, when one considers water availability (NOT precipitation, as they stated) and the processes they are considering, like germination and recruitment. The point has been made earlier the authors need to be clearer about their universe of inference – the statement is not about all plants, but rather forbs and graminoids.

R6> The reviewer correctly points out that we did not explicitly state that this hypothesis was directly linked to studies on productivity of herbaceous communities. We now specify that our hypothesis is founded upon studies on plant productivity both at the plot (focused on plot level surveys), and global scale (L. 84-89, L. 82-87 and L. 103-105, L. 99-101).

L. 84-89, L. 82-87: Assuming plant productivity is a proxy of population performance, we expect that precipitation, or its interaction with temperature, predict herbaceous plant population growth better than temperature alone. Most plant physiological processes, such as seed germination, tissue growth, floral induction, and seed set, are affected by water availability (Jones 2013).

Accordingly, precipitation is the most important driver of vegetation productivity worldwide (Seddon *et al.* 2016).

L. 103-105, L. 99-101: Accordingly, as water availability decreases, precipitation becomes the main factor limiting plant physiological processes (Noy-Meir 1973, Huxman *et al.* 2004).

L. 92: Water availability is NOT the difference between AET and precipitation. Water availability to plants also depends on their rooting zone and depth, as well as soil texture, since clays hold water unavailable more than sands, for instance. I accept the statement as a very broad simplification, but I think the authors should demonstrate that they understand their characterisation is very coarse.

R7> This is an additional good point that we were happy to include in the appropriate section of the methods. (**L. 236-241, L. 212-217**):

We calculated the WAI values at each of our sites by subtracting mean annual potential evapotranspiration from the mean annual precipitation. Such proxy is a coarse measure of plant water availability that ignores information such as soil characteristics, and plant rooting depth. However, our WAI is useful to compare water availability among disparate environments, so that it is often employed in global analyses (Vicente-Serrano *et al.* 2013; Klein *et al.* 2015).

I didn't get a good sense of what comprised a climate anomaly (A in Eq. 3). Yes, that was defined on L. 90, and how computed described L. 178-184. They use gridded monthly values (L. 171), which by their nature would be dampened in terms of anomalies (because they are fitted and smoothed to obtain grids) compared with the real climate data. I suspect that sites in their analysis could also have World Met Organisation-standard data actually measured nearby. Were there any comparisons of actual versus gridded climate anomalies for such instances that would support using gridded data as a reasonable representation? Also, why wasn't mean annual precipitation used in Fig. 1A, which would have been more consistent with mean annual temperature? If precipitation is transformed to the water availability index, then shouldn't temperature be transformed, e.g. cumulative degree days.

R8> We share the concerns of the reviewer regarding our climatic predictors. Below, we explain why we chose to gridded climatic data to calculate climatic anomalies, and WAI and mean annual temperature as predictors to quantify average climatic conditions. See points 1 through 3 below.

1. We now point out that we use gridded climatic data because they are particularly suited to estimate annual climatic means, among others (**L. 196-197, L. 171-172**):

Gridded climatic data are especially suited to estimate annual climatic means (Behnke et al. 2016)

2. Indeed, our initial plan was to calculate growing degree days instead of mean annual temperature. However, we need daily weather data to calculate growing degree days, and we do not know a global, downscaled, gridded daily weather data source. We now state this explicitly in the main text of the methods section (L. 241-245, L. 217-221).

As our proxy of temperature limitation, we use mean annual temperature. While growing degree days would be a more mechanistic measure of temperature limitation (McMaster 1997), this requires daily weather data. However, we could not find a global, downscaled, daily gridded weather dataset to calculate this metric.

3. As for the use of water availability index (WAI), we prefer it to mean annual precipitation (MAP) for two reasons. First, WAI is more linked to mechanisms, just as growing degree days would be (McMaster 1997). Second, in our preliminary analyses, using MAP to test hypothesis 2 produced a clearly significant result, rather than the marginally significant result that we observed performing this test using WAI. Specifically, testing hypothesis H2 produced 96.9% of bootstrapped β_{meta} values below 0 (Figure below), versus the 90.5% of the same test performed using WAI (Fig. 1A in the main text). This clearly significant result obtained using a less mechanistic predictor might in part reflect a spurious correlation. Therefore, we believe using WAI as a predictor is the most conservative choice.

REFERENCES

- McMaster, G. (1997). Growing degree-days: one equation, two interpretations. *Agricultural and Forest Meteorology*, 87, 291–300.
- Morris, W.F., Ehrlén, J., Dahlgren, J.P., Loomis, A.K. & Louthan, A.M. (2020). Biotic and anthropogenic forces rival climatic/abiotic factors in determining global plant population growth and fitness. *PNAS*, 117, 1107–1112.
- Shefferson, R.P., Kull, T., Hutchings, M.J., Selosse, M.-A., Jacquemyn, H., Kellett, K.M., *et al.* (2018). Drivers of vegetative dormancy across herbaceous perennial plant species. *Ecology Letters*, 21, 724–733.

Reviewer #2 (Remarks to the Author):

The main strength of the paper is that provides important information about the effects of variation in temperature and precipitation on projected population growth rates in plants by examining currently available information from long-term demographic studies of plants. I do, however, think that the paper needs to, or can, be improved in several respects.

First, I think that the reasoning regarding the expected relationship between biomes and effects of climatic anomalies – hypotheses 2 and 3 - is a bit simplistic. As argued by the authors, it is reasonable to assume that effects of drought or temperature anomalies will be more severe when conditions are already dry or temperatures high. However, while this is true for a population with a given genetic composition, we might also expect that in arid or high-temperature areas, species which are sensitive to drought or high temperatures might already have been filtered out, or populations might have adapted locally to resist extreme conditions.

R9> This is a fair point, which we now address in the introduction, methods, and discussion:

1. In the introduction, we clarify our expectation that climate anomalies will have more pronounced effects in arid and cold biomes by stating that even if species should be adapted to their environment, our hypotheses posit that in extreme environments, hard physiological limits to plant life are reached more frequently (**L. 95-109**, L. 93-105):

Precipitation and temperature anomalies are expected to have more pronounced effects in arid and cold biomes than in wet and temperate ones. While species should be adapted to their respective environment, extreme environments impose hard physiological limitations more frequently. In arid environments plants experience water limitation more frequently (Knapp et al. 2008). Similarly, in cold biomes plants will more frequently experience temperatures that are too low to allow tissue growth (Alvarez-Uria & Körner 2007; Körner 2008). Accordingly, as water availability decreases, precipitation becomes the main factor limiting plant physiological processes (Noy-Meir 1973, Huxman et al. 2004). On the other hand, in cold biomes temperature anomalies can be disproportionately important. For example, the effect of temperature on tree growth increases with altitude (Galván et al. 2014; Primicia et al. 2015). Similarly, in the tundra temperature anomalies can dramatically change the length of the growing season (Bryson 1974).

2. In the methods, we now more clearly lay out our null hypothesis for hypotheses 2 and 3. We state that our use of standardized climatic anomalies (z-scores) is designed to reflect a null model of perfect to the local climate (**L. 213-224**, L. 189-200). Using z-scores is common practice in global studies of vegetation response to climate (Vicente-Serrano *et al.* 2013; Seddon *et al.* 2016). Moreover, we explicitly explain the significance of our null model (**L. 363-368**, L. 312-317):

L. 213-224, L. 189-200: Z-scores are commonly used in global studies on vegetation responses to climate (Vicente-Serrano et al. 2013; Seddon et al. 2016), and they reflect the null hypothesis that species are adapted to the climatic variation at their respective sites. Across our populations, the

standard deviations of annual precipitation and temperature anomalies change by 300% and 60%, respectively (Fig. S2). Thus, a z-score of one refers to a precipitation anomaly of 50 or 160mm; of 0.5 or 0.8° Celsius. Our null hypothesis posits that species are adapted to these conditions, regardless of the absolute magnitude of the standard deviation in annual climatic anomalies. If this were true, each species would respond similarly to z-scores. However, we found our temperature and precipitation z-scores were highly skewed (skewness above 1) only in respectively two (for temperature) and three (for precipitation) of our 162 populations. We concluded that this degree of skewness should not bias our z-scores substantially.

L. 363-368, L. 312-317. The null hypothesis of these meta-regressions is that plant species are adapted to the climatic variation at their respective sites. Such adaptation implies that a precipitation z-score of one has similar effects regardless the magnitude of the absolute climatic it refers to (Fig. S2). On the other hand, our hypotheses posit that at low WAI and MAT values, species are more responsive to z-scores than expected under the null hypothesis.

3. In the discussion, we address our failure to reject the null hypotheses for hypotheses 2 and 3 by explaining its biological implications, and relevance to climate change vulnerability at the biome level (L. 569-581, L. 442-455):

The fact that responses to climate do not change based on biome suggests that plant populations are demographically adapted to cope with climate variation regardless of the average climate. In extreme environments, the stronger effect of climate on the variation of ecosystem processes such as productivity (Huxman et al. 2004; Chen et al. 2019) or biomass accumulation (Galván et al. 2014; Primicia et al. 2015) is not reflected in demographic patterns. It is therefore plausible that adaptations such as investment in survival (Fridley 2017) or dormancy (Gremer & Venable 2014) are sufficient to de-couple physiological processes from demographic patterns. Such de-coupling is crucial, because if climate drove larger variance in population growth rates, this would decrease the chances of population persistence. However, because plants appear adapted to their local climatic variation, these results do not mean that all biomes will be equally vulnerable to climatic change. Rather, vulnerability to climate change will likely depend on how changes in climate compare to pre-existing climatic variability (Sheldon et al. 2018).

Second, I think that the logic behind the hypothesized relationship between parity and sensitivity to extreme weather is poorly explained and hard to understand. In fact, the authors admit in the discussion that their hypothesis might be wrong. Given this, and that relationships with parity and longevity are examined in separate statistical models although there are good reasons to expect that they are strongly correlated, I think that hypothesis 5 should be dropped.

R10> We believe reviewer 2 is correct, and that removing hypothesis 5 from this article is a sensible and conservative choice. We have therefore removed this hypothesis test from the article.

Thirdly, I think that the part about temporal replication should can be omitted from the paper. This is of two reasons. The first is that I think that this question is conceptually poorly connected to the rest of the study. The second, and most important, reason is that the analyses carried out by the authors does not tell us anything about how temporal replication affects the ability to correctly assess the effects of anomalies in temperature or precipitation on population growth rate, only about how much of the proportion of climatic variation that is sampled. If relationships between climatic factors and population growth rates are linear, which is assumed in all statistical models fitted by the authors, then it is not the range of climatic variation sampled that determines the ability to correctly assess the effects. It is instead the magnitude of temperature effects vs. the effects of other environmental drivers and sampling errors. If relationships can be correctly estimated for a smaller range of climate conditions, then only climate data are necessary to extrapolate effects of more extreme weather conditions as long as relationships are linear.

R11> The two arguments proposed by reviewer 2 convinced us to remove this analysis from the article. We removed this analysis because the range of climatic conditions experienced during a study is only one of the factors affecting statistical power. Specifically, assuming a linear relationship, statistical power depends on total sample size, on the “true” magnitude of climatic effects, and on the range of climatic conditions experienced during the study.

This brings me to my fourth concern; the lack of tests for non-linear relationships and the lack of discussion of the likelihood that effects of climate might be non-linear and that effects of extreme weather conditions cannot always be predicted from relationships under more restricted range of variation. I realize that for many of the species, the temporal replication is too poor to allow for such a test. However, for several species in your data set the number of years is clearly sufficient to test for the presence of non-linear relationships.

R12> We have tested for nonlinear relationships in 38 of total of 162 populations, for which more than 15 years of data were available, finding support for nonlinear relationships in only 8 of these (**L. 437-440, L. 356-359**). Moreover, we have created a new appendix to show scatter plots representing these 8 nonlinear relationships (Fig. S4-6).

L. 437-440, L. 356-359: Our model selection provided some evidence for nonlinear responses to climate, and little evidence of interactions between climatic and non-climatic factors. A nonlinear model was selected in eight of the 38 populations for which we tested nonlinear relationships. In four of these eight populations, nonlinearity in the response was driven almost exclusively by the annual temperature observed between years 1933 and 1934 (Fig. S4-6)

Fifth, a large number of recent studies that have demonstrated that the climate experienced locally by, for example a plant population, might differ dramatically from the regional climate, and that several features of the physical and biotic environment might buffer variation in the regional

climate. Against this background it would be interesting to discuss the limitation of studies, such as this, that are based on gridded temperature data.

R13> We thank reviewer 2 for bringing up a topic that is important and poised to gain increasing attention in the near future. We have added a paragraph to discuss the virtues and limitations of gridded temperature datasets on quantifying climatic effects on plant population growth rates (L. 602-618, L. 476-492):

The predictive ability of our results, which use as predictor annual climatic anomalies calculated from gridded climatic data, could be improved in the future by mechanistic models that use increasingly more available micro-climatic information (Lembrechts & Lenoir 2020). Gridded climatic data are adequate to estimating climatic means registered by weather stations over long time periods, such as years (Behnke et al. 2016). However, the temperature experienced by plant tissues can sometimes be substantially different from the air temperature registered by weather stations (Löffler et al. 2006; Scherrer & Körner 2010). We note, however, that this fact does not invalidate the use of gridded climatic data, because annual anomalies observed at the micro-climatic and weather station level should be similar. For example, Scherrer and Korner (2010) show a tight linear relationship between air temperature and the micro-climate at the leaf surface in alpine vegetation. Nevertheless, micro-climatic data will be required to test mechanistic models of climatic effects, such as those linked to thresholds. Examples of these thresholds are growing degree days (Mcmaster 1997) or frost damage (Lenz et al. 2016). Similarly, micro-climatic anomalies could help understand why different populations of the same species respond differently to similar climatic anomalies (e.g. Nicole' et al. 2011).

Sixth, I think that the authors need to better explain why they consider z-values as biologically relevant measures of climatic variation in this context, and in what respect this measure might be problematic. For example, I imagine that for some populations, yearly values of precipitation and temperature might be quite skewed.

R14> We have now added an explanation of our use, and biological relevance, of z-values in a new paragraph of the methods (L. 213-224, L. 189-200):

Z-scores are commonly used in global studies on vegetation responses to climate (Vicente-Serrano et al. 2013; Seddon et al. 2016), and they reflect the null hypothesis that species are adapted to the climatic variation at their respective sites. Across our populations, the standard deviations of annual precipitation and temperature anomalies change by 300% and 60%, respectively (Fig. S2). Thus, a z-score of one refers to a precipitation anomaly of 50 or 160mm; of 0.5 or 0.8° Celsius. Our null hypothesis posits that species are adapted to these conditions, regardless of the absolute magnitude of the

standard deviation in annual climatic anomalies. If this were true, each species would respond similarly to z-scores. However, we found our temperature and precipitation z-scores were highly skewed (skewness above 1) only in respectively two (for temperature) and three (for precipitation) of our 162 populations. We concluded that this degree of skewness should not bias our z-scores substantially.

Seventh, I think that the authors need to be much more careful when interpreting their results in terms of predicting responses to climate change (e.g. lines 447-8) as increases in standardized anomalies are not necessarily equivalent to predicted changes in climate. For example, mean conditions might often be more important to long-term population growth rate than variance in conditions.

R15> Following this comment, and the comment to lines 442-5, we have re-written the last paragraph of the discussion specifying the context and extent to which our results are relevant to climate vulnerability assessments. We now emphasize the inherent challenges in using observational datasets in forecasts. Specifically, we point out that these challenges apply to forecasting increases in the variance and, most of all, the mean of climatic drivers. Finally, this paragraph also comes with a different, and a more substantial reference support (L. 619-636, L. 493-510):

Our findings on the link between short generation times and climatic sensitivity do not automatically translate into climate vulnerability. The observational nature of our data imposes to interpret our findings in light of two caveats. First, our data did not address several of the concurrent factors that contribute to the effects of climate on populations. These include factors such as density-dependence (Ehrlén & Morris 2015), trophic interactions (Van der Putten et al. 2010), and anthropogenic drivers (Morris et al. 2020). Second, our results are more relevant to changes in climatic variability than changes in climatic means. When predicting the effects of large changes in climatic means, our nonlinear results (Fig. S3-5) show that extrapolation might not be warranted. Besides these two caveats, the conservation literature links short generation times to lower, rather than higher climate vulnerability as indicated by our results (Pearson et al. 2014; Butt & Gallagher 2018). These studies reflect conservation assessments which posit that short generation time should be linked to lower extinction probability (Mace et al. 2008). Short-generation time should also increase the probability of evolutionary rescue (Lynch 1993). However, the advantages provided by short generation time might be overridden by the rapid rates of climate change expected. Weighing the positive and negative effects of generation time will leverage our findings to improve the quality of climate change vulnerability assessments.

More specific comments:

Lines 105-7: It seems hard to believe that the population growth rate of long-lived species (with higher survival elasticity) should be less sensitive to changes in survival than short-lived species (with

higher reproduction elasticity). If you maintain that this statement is correct, then I think it needs further explanation.

R16> This was a mistake in the previous version of the manuscript, and we thank the reviewer for catching it. Our original intention was to talk about increases in the *temporal variation* of vital rates. We have updated the text accordingly (L. 119-121, L. 111-113):

We expect this because the long-run population growth rate of long-lived species responds less strongly to increases in the temporal variation of survival, growth, and reproduction (Morris et al. 2008).

Lines 245-62: It is unclear to me on what sort of logic you base these calculations. What kind of problem does effects of other “confounding” environmental factors pose, and how do you “correct” for it by carrying out the calculations you did? Also, what are the implications of that you accounted for the very tiny proportion of effects of non-climatic drivers that were measured and reported in the studies, but neglect the vast majority of effects that were not measured. Lastly, what did you gain from your adjustments? How much did adjusted values differ from raw means?

R17> Here, we interpret this paragraph as addressing (1) the relevance of confounding factors to our analyses, and the implications of accounting for just a tiny proportion of such factors, (2) how to correct for these confounding factors, and (3) the magnitude of the adjustment.

1. We changed our wording to clarify that our original aim for this analysis was to test whether such confounding factors can substantially interact with climatic signals. If these interactions were a common phenomenon, our climate analyses would be biased because, as the reviewer points out, our studies measure only a small proportion of the potential confounding factors (L. 301-313, L. 274-286).

Finally, we also tested whether non-climatic covariates could bias the effects of climate on $\log(\lambda)$ estimated in our analysis. Such bias, either upwards or downwards, could result in the case non-climatic co-variates interacted with climate. We tested for such interaction in the 21 studies that measured non-climatic factors. For these studies, we fit a linear effect of the non-climatic covariate, and its interaction with one of the two linear climatic anomalies. Thus, including the linear model in Eq. 2, the nonlinear models, and the covariate interaction models, we tested up to six alternative models for each one of our populations (Table S2). We selected the best model according to the Akaike Information Criterion corrected for small sample sizes (AICc, Hurvich & Tsai 1989). We carried out these and subsequent analyses in R version 3.6.1 (R Core Team 2019).

2. Given the above, we mostly tested, rather than corrected for, the influence of non-climatic covariates on our results.
3. However, in two cases, we found substantial interactions with a non-climatic covariate which were selected by AICc. In these two cases, the effect size of temperature increased by 400% and by 16% (L. 440-445, L. 359-363):

Only two populations showed a substantial effect of the interaction between climate anomalies and covariates: our only population of *Astragalus cremnophylax* var. *cremnophylax*, and one of *Dicerandra frutescens* (Table S1). These interactions increased the estimates of the climatic effect by 400% (from 0.001 to 0.052) and decreased it by 16% (from -0.189 to -0.158), respectively.

Lines 348-55 and corresponding parts of the discussion: You use much space to report and discuss this non-significant result, and you also start by describing the effect and only 4 lines lower down tell that it was not significant.

R18> We have now removed this part of the discussion, as we discovered the marginally significant result of the meta-regression testing hypothesis 2 was mostly driven by the effect sizes of graminoid species. We now explain our new results on the effect of removing graminoid species on the meta-regressions (L. 487-497, L. 390-400):

The effect of precipitation ($P < 0.01$), but not temperature ($P=0.97$), changed based on organism type according to the ANOVA tests. Tukey's honestly significant difference test showed a significant difference in the effect of precipitation between herbaceous and graminoid species (Table S3-4, Fig. S6). We therefore re-run separate tests of H2, and H4 excluding the effect sizes of graminoid species. We excluded graminoid species only, because herbaceous species comprised 127 of our 162 populations, so that excluding them would not provide meaningful inferences. In these additional tests discarding graminoid data, H2 was not supported, and H4 was upheld. In H2, the percentage of simulated β_{meta} values lower than zero was 72%, well below the 90.4% of the full dataset (Appendix S5, Figure S7). On the other hand, H4 was upheld, with 100% of β_{meta} below zero (Appendix S5, Figure S8).

And in the discussion, we address our failure to reject the null hypothesis when testing hypotheses 2 and 3 (L. 569-581, L. 442-455):

The fact that responses to climate do not change based on biome suggests that plant populations are demographically adapted to cope with climate variation regardless of the average climate. In extreme environments, the stronger effect of climate on the variation of ecosystem processes such as productivity (Huxman et al. 2004; Chen et al. 2019) or biomass accumulation

(Galván et al. 2014; Primicia et al. 2015) is not reflected in demographic patterns. It is therefore plausible that adaptations such as investment in survival (Fridley 2017) or dormancy (Gremer & Venable 2014) are sufficient to de-couple physiological processes from demographic patterns. Such de-coupling is crucial, because if climate drove larger variance in population growth rates, this would decrease the chances of population persistence. However, because plants appear adapted to their local climatic variation, these results do not mean that all biomes will be equally vulnerable to climatic change. Rather, vulnerability to climate change will likely depend on how changes in climate compare to pre-existing climatic variability (Sheldon et al. 2018).

Lines 442-5: This reasoning is very simplistic and seems to neglect the role of trade-offs in life history evolution.

R19> Because we removed the test of hypothesis 5 from the new manuscript, we also removed this entire paragraph from the discussion.

Lines 480-491: This discussion of the pros and cons of demographic approaches and Species Distribution Models is quite superficial and does not add much to the paper. The discussion lacks references to some recent papers exploring effects of climatic factors on plant population growth rate.

R20> We have re-written the last paragraph of the discussion as part of our answer to the seventh concern of this reviewer (See **R15**). In this new paragraph, we refrain from addressing species distribution models. Rather, we discuss the caveats in our results, and how our results will help climate vulnerability assessments (L. 619-636, L. 493-510).

Our findings on the link between short generation times and climatic sensitivity do not automatically translate into climate vulnerability. The observational nature of our data imposes to interpret our findings in light of two caveats. First, our data did not address several of the concurrent factors that contribute to the effects of climate on populations. These include factors such as density-dependence (Ehrlén & Morris 2015), trophic interactions (Van der Putten et al. 2010), and anthropogenic drivers (Morris et al. 2020). Second, our results are more relevant to changes in climatic variability than changes in climatic means. When predicting the effects of large changes in climatic means, our nonlinear results (Fig. S3-5) show that extrapolation might not be warranted. Besides these two caveats, the conservation literature links short generation times to lower, rather than higher climate vulnerability as indicated by our results (Pearson et al. 2014; Butt & Gallagher 2018). These studies reflect conservation assessments which posit that short generation time should be linked to lower extinction probability (Mace et al. 2008). Short-generation time should also increase the probability of evolutionary rescue

(Lynch 1993). However, the advantages provided by short generation time might be overridden by the rapid rates of climate change expected. Weighing the positive and negative effects of generation time will leverage our findings to improve the quality of climate change vulnerability assessments.

Table S1 lists one 5-year study although the text says that only studies with 6 or more years were included. It also lists climate as a non-climatic factor (*Crypthanta flava*).

R21> First, we thank the reviewer – we removed the 5-year study, which entered this synthesis through a mistake in our code. The removed study, focusing on *Primula farinosa*, ran for exactly 6 years, but due to a complete population crash, these data provided only five population growth rates.

Second, we changed *Crypthanta flava*'s factor to "Rainout shelter". This is technically a climatic driver (i.e. drought), but we included it in our test of non-climatic covariates because it does not refer to climatic variation; rather it modified average water availability.

Reviewer #3 (Remarks to the Author):

Dear Editor,

The manuscript "Short-lived plants have stronger demographic responses to climate" is a synthesis of empirical evidence on plant population responses to climatic changes in temperate biomes. The major strength of the study lies in the analysis of the longest time series of demographic observations available to date in plant demography databases. It is sobering that in the areas of the world the most intensely researched the authors found only 46 studies (62 species and 165 populations) that have monitored populations for at least seven years. Even this monitoring effort, as the authors demonstrate it, is not really enough to fully understand plant population responses to the threats of climate change. While responses sought at the biome-level still prove noisy, we can tap on species' evolutionary history to reveal global signals of climate effects on population performance. This is a very strong message that should incentivise longer term demography research, especially in biomes where such data is barely available. I have three, or perhaps four major concerns about the approaches taken:

1) Hypothesis 2 and Hypothesis 3 refer to effects of temperature and precipitation anomalies on species of cold and arid biomes respectively. It is still not clear how the response variable in the meta-regression is independent of the predictor variables, it would be great if this was explained upfront in the text. Further, I was expecting an interaction term between temperature and precipitation in meta-regression, since the authors suggest already in the introduction that such an interaction should be expected, and in H1 even have this term included. I am also puzzled by Eq. 2 and 3, where I was expecting an autocorrelation term among the explanatory variables, since it is well known in demography that yearly growth rates (any temporal measurements) are likely autocorrelated. Overlooking this term in these two types of model could be at the root of weak climate signals obtained.

R22> Below we address (1) the independence of response and predictor in the meta-regressions, (2) the lack of an interaction term in the single population regressions, and (3) the observations regarding autocorrelation in the explanatory variables, and lambda values.

1. We thank the reviewer for pointing out the confusion regarding the independence between the response and predictor variables in the meta-regressions testing H2 and H3. We have changed our approach to this issue, because we realized that talking about the potential for "spurious correlations" is misleading. We addressed this comment i) spelling out the reasons for our use of z-scores, and ii) explaining our null expectation.

Regarding our use of z-values (L. 213-224, L. 189-200):

Z-scores are commonly used in global studies on vegetation responses to climate (Vicente-Serrano et al. 2013; Seddon et al. 2016), and they reflect the null hypothesis that species are adapted to the climatic variation at their

respective sites. Across our populations, the standard deviations of annual precipitation and temperature anomalies change by 300% and 60%, respectively (Fig. S2). Thus, a z-score of one refers to a precipitation anomaly of 50 or 160mm; of 0.5 or 0.8° Celsius. Our null hypothesis posits that species are adapted to these conditions, regardless of the absolute magnitude of the standard deviation in annual climatic anomalies. If this were true, each species would respond similarly to z-scores. However, we found our temperature and precipitation z-scores were highly skewed (skewness above 1) only in respectively two (for temperature) and three (for precipitation) of our 162 populations. We concluded that this degree of skewness should not bias our z-scores substantially.

Regarding our null hypothesis, we assume that all species are adapted to their local climate variation. Under this null hypothesis, the per-mm and per-degree Celsius effect of precipitation and temperature anomalies, respectively, would change three-fold, and by 60% across our WAI and MAT values, respectively. Our hypotheses simply posit that at the lower values of WAI and MAT, populations would be more responsive to standardized anomalies, regardless of what absolute anomaly they refer to (L. 363-368, L. 312-317):

The null hypothesis of these meta-regressions is that plant species are adapted to the climatic variation at their respective sites. Such adaptation implies that a precipitation z-score of one has similar effects regardless the magnitude of the absolute climatic it refers to (Fig. S2). On the other hand, our hypotheses posit that at low WAI and MAT values, species are more responsive to z-scores than expected under the null hypothesis.

Finally, here we note that the “spurious correlation” we mentioned in our previous version of the manuscript makes the implausible assumption that species respond similarly to a per-mm and per-degree Celsius across WAI and MAT values. This is clearly implausible because, using precipitation as an example, if this were true, the effect of precipitation at our highest WAI sites would be three times larger on average. We therefore decided to remove the previous appendix S2 on the “spurious correlation”, and to explicitly describe our null expectation for hypotheses 2 and 3 (see passage immediately above referring to L. 363-368, L. 312-317).

2. We did not include an interaction term in the single-population analyses, because perhaps due to overfitting, we did not find any support for such interaction in a preliminary model selection. In only one out of our 162 populations an interaction term was supported over a multiple regression including both precipitation and temperature anomalies (data not shown).

3. This comment refers to the autocorrelations in both the response variable (λ) and among the explanatory variables. We address the issue of autocorrelation in the response variable, below, responding to the second point raised by reviewer 3, which is also focused on the autocorrelation in the response variable (λ).

As for the autocorrelation among the explanatory variables, we assume here that the reviewer is pointing out that precipitation and temperature anomalies might be correlated through time. We assume this because we do not know of regression designs that account for the autocorrelation in the predictors. On the other hand, if two explanatory variables are strongly correlated, their effect should be estimated using a multiple regression to avoid bias (Freckleton 2011). In our dataset, we found that precipitation and temperature anomalies are mostly correlated negatively, and in 16 populations this correlation is below -0.4. We therefore fit multiple regressions including both temperature and precipitation anomalies (L. 271-275, L. 248-252, Eq. 2). We note, however, that this change did not modify the inferences of our analysis.

L. 271-275, L. 248-252: For each population, we fit multiple regressions with an autoregressive error term, and we evaluated the potential for nonlinear effects in the datasets longer than 14 years. We fit multiple regressions because temperature and precipitation anomalies were negatively correlated, so that fitting separate models for temperature and precipitation would yield biased results (Freckleton 2011).

Updated Equation 2:

$$\log(\lambda)_y = \alpha + \beta_p P_y + \beta_t T_y + \varepsilon_y, \quad \text{Eq. (2a)}$$

$$\varepsilon_y = \rho \varepsilon_{y-1} + \eta_y \quad \text{Eq. (2b)}$$

2) Yearly autocorrelation has also been overlooked when calculating the response variable, lambda. It is well known in demography that population growth rate can differ when calculated with or without accounting for the effect of (mostly) climate autocorrelation on population performance.

R23> This turned out to be an insightful comment which changed some of our model selection results (see below). We re-fit our models using an autoregressive error term of order one ("AR models", Eq. 2 above). However, this did not dramatically change the outcome of our hypothesis tests.

3) While the population growth rate is indeed the most widely used population measure, it is well known in demography that populations are able to buffer both the temporal and the spatial variation in its vital rates. While we surely wish first and foremost to obtain signals from this integrated population measure, existing evidence makes me suspect that more sensitive responses, of the vital rates, have remained hindered in this dataset. I think there would be purpose for a couple of more detailed analyses at least for H2 and H3, where obviously there is need for more in depth explorations.

R24> This is a cogent suggestion, which we considered at an advance stage of this study, but which we decided to address in a separate article. Analyses at the level of vital rates open up substantial conceptual and methodological challenges. Conceptually, the response of vital rates to climate only makes sense in light of buffering theory, which has a long history, and is still controversial in certain regards (McDonald et al. 2017; Hilde et al. 2020). Methodologically, the effect of climate on vital rates need be estimated using beta and gamma regressions, which impose a set of nontrivial challenges. Given the above considerations, we believe that the vital rate analyses warrant a separate, dedicated manuscript.

4) Plant responses even to climate anomalies are not likely to be linear. I appreciate that the authors at least tried to include a quadratic term when testing for H1, and that in Eq 2 and 3 a quadratic term would have been again problematic for most populations. However, would it be possible to look for any evidence in the data for such a non-linear relationship, perhaps in the populations studied over the longest term?

R25> After addressing this comment, and the comment on auto-regressive error terms, we found some support for our nonlinear relationships in datasets of at least 15 years. As the reviewer noted, it makes sense to include nonlinear error terms in sufficiently long datasets. We chose 15 years as a compromise threshold value following Jenkins and Quintana-Ascencio (2020) who, when performing model selection recommend $N = 8$ for datasets with low variance, but a minimum of 25 points for datasets with high variance. We now present these nonlinear models in the methods (L. 294-300, L. 267-273; L. 314-322, L. 287-295), and results (L. 437-440, L. 356-359).

L. 294-300, L. 267-273: We compared the baseline model in Eq. 2 to models including a quadratic climatic effect and non-climatic covariates. We estimated quadratic climatic effects only for time series longer than 14 years. We choose this threshold because when using a model selection approach to select a quadratic or linear regression model, the recommended minimum sample size is between eight and 25 data points (Jenkins & Quintana-Ascencio 2020). We fit models including a quadratic effect of temperature, precipitation, or both (Table S2).

L. 314-322, L. 287-295: In the populations for which AICc selected one of the model alternatives to the baseline in Eq. 2, we calculated the effect size of climate by adding the effect of the new terms to the linear climatic terms. For example, when a quadratic precipitation model was selected, we calculated the effect size of precipitation as $\beta = \beta_1 + \beta_2$. For models including an interaction between temperature and a non-climatic covariate, we evaluated the effect of the interaction at the mean value of the covariate. Therefore, we calculated the effect size as $\beta = \beta_t + \beta_x E(C_i)$ for continuous covariates and $\beta_p + \beta_x 0.5$ for categorical variables. We quantified the standard error of the resulting effect sizes by adding the standard errors of the two terms.

L. 437-440, L. 356-359: Our model selection provided little evidence for nonlinear responses to climate, and little evidence of interactions between

climatic and non-climatic factors. A nonlinear model was selected in eight of the 38 populations for which we tested nonlinear relationships (Fig. S4-6).

Further minor notes:

L85: It is still a puzzle to read this generalisation in this context. Plants do have temperature and drought tolerance curves, and in the context it is sensitive to generalise about how temperature modulates water availability equally across biomes and life histories.

R26> We are not sure we understand the suggestion here. In our introduction we state that **“Temperature can also influence these processes, but typically by modulating water availability (Aparecido et al. 2020), as plant growth occurs across a wide range of temperatures (namely between 5° to 40° Celsius; Körner 2008, Jones 2013)” (L. 89-92, L. 97-90).** We now provide a new and more recent reference in this sentence, Aparecido *et al.* 2020. If possible, we would like to request that the reviewer please provide some ideas of references so we can appropriately modify this statement.

L108: It would be welcome at this point to learn more about the findings in Demetrius 1978.

R27> We have removed this line, as we decided to remove from the study the hypothesis involving the degree of iteroparity, as suggested by reviewer 2.

L172: Is year 1901 a mistake?

R28> This is not a mistake: the CRU TS 4.01 data goes back to year 1901.

L174: I don't agree with this way of calculating the mean monthly temperature, it should be correctly calculated from daily values.

R29> We agree with the reviewer here, but we could not find global, small scale (1Km), downscaled, daily weather data. Hence, we produced this metric using the available gridded climatic data. In the methods, we now state that these data are well suited to estimate annual climatic means (L. 196-197, L. 171-172):

Gridded climatic data are especially suited to estimate annual climatic means (Behnke et al. 2016)

L410: What is the limiting factor then?

R30> We have removed this paragraph, and condensed its meaning in a paragraph on the importance of our geographic and taxonomic biases (L. 582-601, L. 456-475):

The geographic and taxonomic bias of our dataset might amplify the relevance of precipitation anomalies, and it therefore might affect the generality of our findings. First, our geographic bias potentially underemphasizes the role of temperature, because our dataset under-samples extremely cold and hot biomes. For example, in cold biomes such as montane or boreal forests, the influence of temperature on growth is larger as mean annual temperature decreases (Galván et al. 2014; Primicia et al. 2015). On the other hand, in hot biomes, the interaction between precipitation and temperature should become larger than in colder biomes (Aparecido et al. 2020). Therefore, we might expect a strong interaction between precipitation and temperature anomalies where mean precipitation is low and mean temperature high. These conditions should occur, for example, in subtropical desert or tropical savannas; however only a handful of available demographic studies occur in these biomes (Fig. S1).

L475: In Csergo et al. 2017 there is also a great deal of agreement on that the population growth rate is often not the perfect measure of plant responses to climate or habitat suitability. This sentence is best softened, since similar types of disagreement may be envisaged with respect to population responses to climate (or the way we calculate climate variables). Having said these, I agree that generalisations on climate sensitivity directly are highly needed and open up new avenues in comparative demography research.

R31> This comment, as well as the seventh concern by reviewer 2 (see **R15**), convinced us to re-write this last paragraph of the discussion. The new paragraph focuses on the strengths and limitations of our results when informing climate change vulnerability assessments (L. 619-636, L. 493-510):

Our findings on the link between short generation times and climatic sensitivity do not automatically translate into climate vulnerability. The observational nature of our data imposes to interpret our findings in light of two caveats. First, our data did not address several of the concurrent factors that contribute to the effects of climate on populations. These include factors such as density-dependence (Ehrlén & Morris 2015), trophic interactions (Van der Putten et al. 2010), and anthropogenic drivers (Morris et al. 2020). Second, our results are more relevant to changes in climatic variability than changes in climatic means. When predicting the effects of large changes in climatic means, our nonlinear results (Fig. S3-5) show that extrapolation might not be warranted. Besides these two caveats, the conservation literature links short generation times to lower, rather than higher climate vulnerability as indicated by our results (Pearson et al. 2014; Butt & Gallagher 2018). These studies

reflect conservation assessments which posit that short generation time should be linked to lower extinction probability (Mace et al. 2008). Short-generation time should also increase the probability of evolutionary rescue (Lynch 1993). However, the advantages provided by short generation time might be overridden by the rapid rates of climate change expected. Weighing the positive and negative effects of generation time will leverage our findings to improve the quality of climate change vulnerability assessments.

REFERENCES

- Aparecido, L.M.T., Woo, S., Suazo, C., Hultine, K.R. & Blonder, B. (2020). High water use in desert plants exposed to extreme heat, 12.
- Freckleton, R.P. (2011). Dealing with collinearity in behavioural and ecological data: model averaging and the problems of measurement error. *Behav Ecol Sociobiol*, 65, 91–101.
- Hilde, C.H., Gamelon, M., Sæther, B.-E., Gaillard, J.-M., Yoccoz, N.G. & Pélabon, C. (2020). The Demographic Buffering Hypothesis: Evidence and Challenges. *Trends in Ecology & Evolution*, 35, 523–538.
- Jenkins, D.G. & Quintana-Ascencio, P.F. (2020). A solution to minimum sample size for regressions. *PLoS ONE*, 15, e0229345.
- McDonald, J.L., Franco, M., Townley, S., Ezard, T.H.G., Jelbert, K. & Hodgson, D.J. (2017). Divergent demographic strategies of plants in variable environments. *Nat Ecol Evol*, 1, 0029.

REVIEWERS' COMMENTS

Reviewer #1 (Remarks to the Author):

This was the second time this manuscript was reviewed, and it was great to reach something so much improved by responses to previous reviewers. I appreciate the changes made by the authors and in particular feel that the clarification of drought/water availability and temperature anomalies was a nice improvement. I offer two minor comments at this point.

1) The authors' response to Reviewer #1 was that:

"We do not mention the term "herbaceous" in the title, because our dataset contains several perennial plants types, including woody species, succulents, ferns, and graminoids."

I'm afraid that I must push back a little regarding the term 'herbaceous'. Herbaceous plants are: 'Applied to plants which do not develop wood in the stem or branches' (Oxford English Dictionary online). This includes graminoids, forbs, and ferns, and succulents lacking a woody stem. To me, this best describes the spirit of the dataset these authors have assembled (62 species analysed, of which there were 2 trees and 5 shrubs, e.g., 89% of them were herbaceous under the OED definition). I leave this up to the editor, but I think they should include herbaceous in the title.

In lines 141-143, I recommend wording such as:

"Specifically, among our 62 species, this dataset contains 55 herbaceous perennials (11 of which were graminoids and one of which was a fern), five shrubs and two trees."

2) I would suggest a softer take such as "precipitation is a key important driver of vegetation productivity worldwide", rather than "the most important driver ... worldwide". For every statement about the importance of precipitation and drought for global productivity such as Zhao and Running (2010; *Science* v. 329), there are also analyses involving temperature or those contending that moisture availability to plants is intertwined with temperature such as Park Williams et al. 2013 *Nature Climate Change* 3, 292–297. Rather than entering into the controversy, why don't the authors simply bypass the fray and just say that precipitation is important? Does it have to be *most important*?

Reviewer #2 (Remarks to the Author):

I have now read the revised version of the manuscript. Overall, I find that the reviewers have addressed my comments in a good way and that the revised version is much clearer and more focused. I have no major concerns with the revised version and think that it will constitute a valuable contribution to our knowledge about how climatic variation affects plant population dynamics.

One thing that is still not clear to me is why the authors assume that non-climatic covariates should bias the estimates of effects of climate. I can understand why non-climatic covariates interacting with climate add noise but it is not obvious why they should bias estimates. It would be good if the authors could clarify their reasoning here.

A minor thing in this context is the calculations of the corrected effect sizes (line 293) – why should the weight for the categorical variables be 0.5 rather than the relative frequency?

One additional question arose while reading the response to one of the comments by reviewer 1 (R5). In their response, the authors write "... we performed a systematic literature review to identify and digitize the most recent demographic studies linking plant demography to climatic drivers." This made me wonder why you restricted the search to studies linking plant demography to climatic variables as you anyway extracted information about climate from other sources that

the original study. Did this mean that you might have missed some long-term studies not linking plant demography to climate variables? Also, why is not this information provided in the manuscript?

Reviewer #3 (Remarks to the Author):

Dear Editor,

The authors present a much clearer and distilled version of the manuscript, which I believe is very valuable. While probably longer-term observations for a larger number of species (not yet available to science) would likely produce more comprehensive assessments, this work is eye opener for current limitations, and opportunities, in forecasts of population responses to climate conditions.

I also highly value the efforts of authors to improve their approaches in response to reviewer comments and they have clarified all my concerns throughout the response to all reviewers.

However I still feel the authors do not fundament convincingly theoretical expectations about biomes, and the link between biomes/climate, life-history and types of plants (newly added to the revised ms) has not been made. Instead, these look like separate analyses. Certain life-history strategies are better represented in different biomes and as a result, a representative sample of those strategies would be needed/suffice to characterise climate responses in those biomes. This could be a hook, - and an opportunity to talk about the importance of examining the climate responses of different growth forms, functional groups, (or plant types), which is missing from the introduction.

Other than this note, I only have a few minor comments left, detailed below:

In several places, the wording is still misleading, superficial or lacks the elegance of scientific language, e.g.:

L97: "will" suggests a hypothesis to me and it is misleading this way, I suggest using present tense

L102 and L 104: I welcome these example but I feel they is still not conveying sufficient information to the reader, e.g. what magnitude of effects should we think of, and what is the direction of the effects?

L194: I would replace ";" with " and to a temperature anomaly of".

L239: "remaining": I think there were only four hypotheses altogether. "meta-regression": mentioned as early as here, and in this way, to me this is very misleading, because the chapter is presenting completely different types of analyses (needed for the meta-regressions presented in a different chapter)

L299: Primarily the study is not focused on the correlation between these two limiting factors.

L306: "Biome should affect": Of course the authors think about the climate conditions that result in specific biomes (because biome is the community of plants and animals characteristic to a specific environment, so, formulated this way the reader thinks of tests of biotic interactions).

L314 and 317: I welcome these clarifications, but can the z-score have an effect? Are species responsive to z-score, since the z-score is quantifying a response on itself?

L315: sentence incomplete. Figure S2 refers not to z-scores, but standard deviations, yet the text suggests so.

L333: A clarification of why gamma-regressions were used is missing.

L344: Very surprising to learn about plant types for the first time in the Methods section.

L349: Difference tests on the differences, this should be better clarified.

L365 and the whole paragraph: We have little opportunity in this ms and Supporting material to see plots, summary plots, or tables of these very interesting results. I would be very happy to see the results for all species in the supporting material, in addition to the very nice plots showing the

quadratic effects.

L391: "based on organism type": sloppy wording

Discussion: maybe I missed it, but a mention about why grasses are so different in their climate responses is missing.

In sum, I congratulate the authors for their work, and I am looking forward to seeing it published

We provide our responses point by point in the format “**RX**”, where **X** is the response number. In these responses, line numbers in blue (e.g. L. 315-317) refer to the document where changes have been accepted; line numbers in orange (L. 367-369) refer to the document with track changes on.

REVIEWERS' COMMENTS

Reviewer #1 (Remarks to the Author):

This was the second time this manuscript was reviewed, and it was great to reach something so much improved by responses to previous reviewers. I appreciate the changes made by the authors and in particular feel that the clarification of drought/water availability and temperature anomalies was a nice improvement. I offer two minor comments at this point.

R1> We thank reviewer 1 for the positive assessment of our work!

1) The authors' response to Reviewer #1 was that:

"We do not mention the term “herbaceous” in the title, because our dataset contains several perennial plants types, including woody species, succulents, ferns, and graminoids."

I'm afraid that I must push back a little regarding the term 'herbaceous'. Herbaceous plants are: 'Applied to plants which do not develop wood in the stem or branches' (Oxford English Dictionary online). This includes graminoids, forbs, and ferns, and succulents lacking a woody stem. To me, this best describes the spirit of the dataset these authors have assembled (62 species analysed, of which there were 2 trees and 5 shrubs, e.g., 89% of them were herbaceous under the OED definition). I leave this up to the editor, but I think they should include herbaceous in the title.

R2> We changed the title to “Herbaceous perennial plants with short generation time have stronger responses to climate anomalies than those with longer generation time”.

In lines 141-143, I recommend wording such as:

"Specifically, among our 62 species, this dataset contains 55 herbaceous perennials (11 of which were graminoids and one of which was a fern), five shrubs and two trees."

R3> Thanks to this suggestion, we changed this sentence to (L. 315-317; L. 367-369):

Specifically, among our 62 species, this dataset contains 54 herbaceous perennials (11 of which graminoids), and eight woody species: five shrubs, two trees, and one woody succulent (Opuntia imbricata).

2) I would suggest a softer take such as "precipitation is a key important driver of vegetation productivity worldwide", rather than "the most important driver ... worldwide". For every statement

about the importance of precipitation and drought for global productivity such as Zhao and Running (2010; Science v. 329), there are also analyses involving temperature or those contending that moisture availability to plants is intertwined with temperature such as Park Williams et al. 2013 Nature Climate Change 3, 292–297. Rather than entering into the controversy, why don't the authors simply bypass the fray and just say that precipitation is important? Does it have to be *most important*?

R4> We are happy to implement this suggestion, which we agree clarifies an important point. We have changed this sentence to (L. 84, L. 86-87):

precipitation is a key driver of vegetation productivity worldwide

Reviewer #2 (Remarks to the Author):

I have now read the revised version of the manuscript. Overall, I find that the reviewers have addressed my comments in a good way and that the revised version is much clearer and more focused. I have no major concerns with the revised version and think that it will constitute a valuable contribution to our knowledge about how climatic variation affects plant population dynamics.

R5> We thank reviewer 2 for this encouraging evaluation!

One thing that is still not clear to me is why the authors assume that non-climatic covariates should bias the estimates of effects of climate. I can understand why non-climatic covariates interacting with climate add noise but it is not obvious why they should bias estimates. It would be good if the authors could clarify their reasoning here.

R6> We believe that this issue arose because our original text lacked a specific example. We now included a new sentence providing such example (L. 445-447; L. 466-468):

For example, harvest can have multiplicative, rather than additive effects on the climate responses of forest understory herbs⁷⁴.

A minor thing in this context is the calculations of the corrected effect sizes (line 293) – why should the weight for the categorical variables be 0.5 rather than the relative frequency?

R7> This is another fair point of confusion, which we appreciate the reviewer for shining more light on. We modified the end of the sentence to specify that our aim was to use an average effect size between the two categories (L. 463-466; L. 484-487):

Therefore, we calculated the effect size as $\beta = \beta_t + \beta_x E(C_i)$ for continuous covariates. For categorical variables, we calculated the effect size as $\beta_p + \beta_x 0.5$: that is, we calculated the mean effect size between the two categories.

Here, we have referred to categorical variables with two levels to keep the text succinct. While we did have five studies with categorical variables that had more than two categorical levels, we did not find support for a significant interaction in any of them.

One additional question arose while reading the response to one of the comments by reviewer 1 (R5). In their response, the authors write “... we performed a systematic literature review to identify and digitize the most recent demographic studies linking plant demography to climatic drivers.” This made me wonder why you restricted the search to studies linking plant demography to climatic variables as you anyway extracted information about climate from other sources that the original study. Did this mean that you might have missed some long-term studies not linking plant demography to climate variables? Also, why is not this information provided in the manuscript?

R8> We thank the reviewer for pointing this out. We have now modified the first three paragraphs of the methods by including a description of this literature review to improve clarity (L 286-309; L. 307-330; see below).

Regarding the possibility of missing long-term studies, we carried out this literature review because studies linking climatic drivers to plant demography preferentially use long-term demographic data. Accordingly, our review selected the Chu et al. (2016) dataset, which to our knowledge contains the most temporally and taxonomically replicated monitoring demographic dataset on perennial herbaceous species in the world.

To address our hypotheses, we used matrix population models (MPMs) or Integral projection models (IPMs) from the COMPADRE Plant Matrix Database (v. 5.0.156) and the PADRINO IPM Database⁵⁷, which we amended with a systematic literature search. First, we selected density-independent models from COMPADRE and PADRINO which described the transition of a population from one year to the next. Among these, we selected studies with at least six annual transition matrices, to balance the needs of adequate yearly temporal replicates and sufficient sample size of data for quantitative synthesis. This yielded data from 48 species and 144 populations.

We then performed a systematic literature search for studies linking climate drivers to structured population models in the form of either MPMs or IPMs. We performed this search on ISI Web of Science for studies published between 1997 and 2017. We used a Boolean expression containing key words related to plant form, structured demographic models, and environmental drivers (Supplementary Methods). We only considered studies linking macro-climatic drivers to natural populations (e.g. transplant experiments and studies focused on local climatic factors such as soil moisture, light due to tree fall gaps, etc. were excluded). Finally, we used the same criteria used to filter studies in COMPARE and PARDINO, by selecting studies with at least six, density-independent, annual projection models. This search brought two additional species, belonging to three additional populations, which we entered in the COMPADRE database.

One of the studies we excluded from the literature search because it contained density-dependent IPMs, also provided raw data with high temporal replication (14 to 32 years of sampling) for 12 species from 15 populations⁵⁸. Therefore, we re-analyzed these freely available data to produce density-independent MPMs that were directly comparable to the other studies in our dataset (Supplementary Methods).

Reviewer #3 (Remarks to the Author):

Dear Editor,

The authors present a much clearer and distilled version of the manuscript, which I believe is very valuable. While probably longer-term observations for a larger number of species (not yet available to science) would likely produce more comprehensive assessments, this work is eye opener for current limitations, and opportunities, in forecasts of population responses to climate conditions. I also highly value the efforts of authors to improve their approaches in response to reviewer comments and they have clarified all my concerns throughout the response to all reviewers.

However I still feel the authors do not fundament convincingly theoretical expectations about biomes, and the link between biomes/climate, life-history and types of plants (newly added to the revised ms) has not been made. Instead, these look like separate analyses. Certain life-history strategies are better represented in different biomes and as a result, a representative sample of those strategies would be needed/suffice to characterise climate responses in those biomes. This could be a hook, - and an opportunity to talk about the importance of examining the climate responses of different growth forms, functional groups, (or plant types), which is missing from the introduction.

R9> This concern raises a fascinating and highly relevant consideration that it is rather challenging - or even impossible – to disentangle the effect of biome and plant type on responses to climate, given how climate itself also filters plant types (e.g. succulents are more often found in drylands, Hutchinson 2019). This challenge arises because each biome will be dominated by a different plant type, and as we showed, each plant type could in turn have different responses to climate. We first point out this issue in the introduction, at the end of the third paragraph (L. 101-104; L. 106-109):

However, because plant functional composition is filtered by biome¹⁸, it is important to consider whether differences in the responses of plants across biomes might be due to the different composition of plant functional types (graminoids, herbs, ferns, woody species, and succulents) that occur in those biomes.

Accordingly, we then address this topic with a new paragraph of the discussion. Importantly our grasses, which respond to precipitation differently than the other plant types, all occur in arid biomes. In the discussion, we address this issue in the following paragraph (L. 241-250; L. 258-267):

Our data on graminoids exemplify that the covariation between taxonomies and biomes complicates the interpretation of global comparative studies. In our results, the response of graminoids to precipitation anomalies is larger than other plant types, and this response drives the positive correlation between WAI and the effect of precipitation (Fig. 1A). Moderately arid climates favor grasses⁴³, which might have an inherent advantage in exploiting precipitation, or at least precipitation pulses that increase the moisture of shallow soil horizons¹¹. As a result, we cannot establish whether sensitivity to precipitation anomalies is characteristic of graminoids, or, as we originally expected (H2), of arid biomes. In future studies, disentangling the role of biomes and taxonomic bias on plant climate sensitivity will require study designs that stratify plant types across biomes.

Other than this note, I only have a few minor comments left, detailed below:

In several places, the wording is still misleading, superficial or lacks the elegance of scientific language, e.g.:

L97: "will" suggests a hypothesis to me and it is misleading this way, I suggest using present tense

R10> We changed "will" to "should" (L. 94; L. 97)

L102 and L 104: I welcome these example but I feel they is still not conveying sufficient information to the reader, e.g. what magnitude of effects should we think of, and what is the direction of the effects?

R11> We thank the reviewer for soliciting a more precise statement, which we made pointing out that (L. 98-99; L. 102-104):

For example, temperature has a positive effect on tree growth that increases in explanatory power with altitude^{15,16}

L194: I would replace ";" with " and to a temperature anomaly of".

R12> We made this change (L. 365; L. 386).

L239: "remaining": I think there were only four hypotheses altogether. "meta-regression": mentioned as early as here, and in this way, to me this is very misleading, because the chapter is presenting completely different types of analyses (needed for the meta-regressions presented in a different chapter)

R13> Thank you for catching this typo; we changed this sentence to (L. 408; L. 429):

To test our remaining three hypotheses, we carried out meta-regressions

L299: Primarily the study is not focused on the correlation between these two limiting factors.

R14> We now realize that our wording was confusing. We have changed it to (L. 470-472; L. 491-493):

We used a simulation procedure to run two meta-regressions to test for the correlation between the effect size of climate drivers on λ , and our measures of water or temperature limitation.

L306: "Biome should affect": Of course the authors think about the climate conditions that result in specific biomes (because biome is the community of plants and animals characteristic to a specific environment, so, formulated this way the reader thinks of tests of biotic interactions).

R15> We changed this wording to unambiguously address our hypotheses (L. 478-480; L. 499-501):

To test H2 and H3 on how water and temperature limitation should affect the response of populations to climate, we used linear meta-regressions

L314 and 317: I welcome these clarifications, but can the z-score have an effect? Are species responsive to z-score, since the z-score is quantifying a response on itself?

R16> We could not fully understand these two questions, but we addressed them as best as we could. We now clearly point out that we do expect an effect of z-scores, but that we expect this effect not to change across sites with different climatology (L. 486-488; L. 507-509):

Such adaptation implies that a precipitation z-score of one should produce effects on $\log(\lambda)$ of similar magnitude and sign across different climates.

L315: sentence incomplete. Figure S2 refers not to z-scores, but standard deviations, yet the text suggests so.

R17> Thank you for this observation! We have changed this sentence, splitting it in two (L. 486-489, L. 507-510) also see response immediately above):

Such adaptation implies that a precipitation z-score of one should produce effects on $\log(\lambda)$ of similar magnitude and sign across different climates. This should happen across average climatic values that are connected to substantially different absolute climatic anomalies (Supplementary Figure 2).

L333: A clarification of why gamma-regressions were used is missing.

R18> We changed the wording of this section to emphasize that we used gamma regressions to accommodate the support of our response variable, which is bounded between zero and infinity (L. 506-511; L. 527-532):

To test H4 on how the generation time of a species should mediate its responses to climate, we used a gamma meta-regression. We fitted gamma meta-regressions because our response variables were the absolute effect sizes of precipitation and temperature anomalies, $|\beta|$, which are bounded between 0 and infinity. To test H4, we therefore fit gamma meta-regressions with a log link, using $|\beta|$ values as response variable and generation time (T) as predictor.

L344: Very surprising to learn about plant types for the first time in the Methods section.

R19> As explained in **R9**, we have modified the introduction to address this issue which should be common to most comparative global studies (L. 101-104; L. 106-109):

However, because plant functional composition is filtered by biome¹⁸, it is important to consider whether differences in the responses of plants across biomes might be due to the different composition of plant functional types (graminoids, herbs, ferns, woody species, and succulents) that occur in those biomes.

L349: Difference tests on the differences, this should be better clarified.

R20> We acknowledge that this wording was not very elegant and somewhat confusing. We changed the sentences to (L. 520-523; L. 541-544):

We ran ANOVA tests to verify whether the effect sizes of precipitation and temperature anomalies differed between plant types. We then tested for significant differences in pairwise contrasts between plants types by running Tukey's honestly significant difference tests.

L365 and the whole paragraph: We have little opportunity in this ms and Supporting material to see plots, summary plots, or tables of these very interesting results. I would be very happy to see the results for all species in the supporting material, in addition to the very nice plots showing the quadratic effects.

R21> We enthusiastically implemented this suggestion, which dramatically improved the transparency of our analysis. We added these plots as automatically generated PDFs in the github repository that contains our data and code for the analyses of this article (<http://doi.org/10.5281/zenodo.4516446>).

L391: "based on organism type": sloppy wording

R22> We changed this to **"plant type"**.

Discussion: maybe I missed it, but a mention about why grasses are so different in their climate responses is missing.

R23> We have addressed this in the response to reviewer 3's very first concern (**R9**). In short, we hypothesize that in moderately arid climates, grasses might have an advantage in exploiting precipitation pulses that recharge shallow soil horizons.

In sum, I congratulate the authors for their work, and I am looking forward to seeing it published.
Best wishes,
Anna Maria Csergo

REFERENCES

Hutchinson, C. (2019). *Arid Lands: Today And Tomorrow*. Routledge.